# Evolution of natural lifespan variation and molecular strategies of extended lifespan in yeast

Alaattin Kaya[1]*, Cheryl Zi Jin Phua[2], Mitchell Lee[3], Lu Wang[4], Alexander Tyshkovskiy[5,6], Siming Ma[2], Benjamin Barre[5], Weiqiang Liu[7], Benjamin R Harrison[3], Xiaqing Zhao[3], Xuming Zhou[5], Brian M Wasko[8], Theo K Bammler[4], Daniel EL Promislow[3,9], Matt Kaeberlein[3], Vadim N Gladyshev[5]*

[1]Department of Biology, Virginia Commonwealth University, Richmond, United States; [2]Genetics, Genome Institute of Singapore, Agency for Science, Technology and Research (A*STAR), Singapore, Singapore; [3]Department of Laboratory Medicine and Pathology, University of Washington, Seattle, United States; [4]Department of Environmental and Occupational Health Sciences, University of Washington, Seattle, United States; [5]Division of Genetics, Department of Medicine, Brigham and Women's Hospital, Harvard Medical School, Boston, United States; [6]Belozersky Institute of Physico-Chemical Biology, Moscow State University, Moscow, Russian Federation; [7]Key Laboratory of Animal Ecology and Conservation Biology, Chinese Academy of Sciences, Institute of Zoology, Beijing, China; [8]Department of Biology, University of Houston - Clear Lake, Houston, United States; [9]Department of Biology, University of Washington, Seattle, United States

*For correspondence:
kayaa@vcu.edu (AK);
vgladyshev@rics.bwh.harvard.edu (VNG)

**Abstract** To understand the genetic basis and selective forces acting on longevity, it is useful to examine lifespan variation among closely related species, or ecologically diverse isolates of the same species, within a controlled environment. In particular, this approach may lead to understanding mechanisms underlying natural variation in lifespan. Here, we analyzed 76 ecologically diverse wild yeast isolates and discovered a wide diversity of replicative lifespan (RLS). Phylogenetic analyses pointed to genes and environmental factors that strongly interact to modulate the observed aging patterns. We then identified genetic networks causally associated with natural variation in RLS across wild yeast isolates, as well as genes, metabolites, and pathways, many of which have never been associated with yeast lifespan in laboratory settings. In addition, a combined analysis of lifespan-associated metabolic and transcriptomic changes revealed unique adaptations to interconnected amino acid biosynthesis, glutamate metabolism, and mitochondrial function in long-lived strains. Overall, our multiomic and lifespan analyses across diverse isolates of the same species shows how gene–environment interactions shape cellular processes involved in phenotypic variation such as lifespan.

## Editor's evaluation

This manuscript characterizes differences in replicative lifespan across a collection of natural yeast isolates. Additionally, it provides transcriptional and metabolomic resources that help shed important new light on the mechanistic bases of functional differences among the wild yeast isolates. The revised version further tests the impact on lifespan in short- and long-lived strains of target genes identified. This work has important implications for the fields biology of aging and potentially evolutionary biology.

## Introduction

Diverse selective forces (mutation, selection, and drift) generate enormous variation within and among species (*Via and Lande, 1985*; *Li et al., 2018*). Consequently, many morphological, behavioral, and physiological phenotypes (traits) vary within and between species in natural populations (*Wagner and Zhang, 2011*; *Sambandan et al., 2008*; *Malagón et al., 2014*; *Caswell, 2007*; *Fusco and Minelli, 2010*). For example, genetic variation in natural populations of many organisms can differentially affect their neural and endocrine functions, leading to variation in quantitative life-history traits such as fitness and age at maturation (*Bonier et al., 2009*; *Finch and Rose, 1995*). Variation in another fitness trait, lifespan, has also attracted much attention (*Arking et al., 1996*; *Libert and Pletcher, 2007*; *Ratikainen and Kokko, 2019*). Across eukaryotic species, longevity can differ over many orders of magnitude, from days to centuries, for example the Greenland shark (*Somniosus microcephalus*) may live for more than 500 years whereas some species live only several days (*Mortimer and Johnston, 1959*; *Nielsen et al., 2016*; *Jones et al., 2014*). Longevity also varies among individuals of the same species, indicating that variability of lifespan is not constrained at the level of species, and that the molecular determinants of lifespan vary within the same genetic pool (*Finch and Pike, 1996*; *Klass, 1983*; *Kaya et al., 2015*; *Ma et al., 2018*; *Dato et al., 2018*; *Wright et al., 2019*).

What, then, are the factors that determine the lifespan of individuals? The molecular pathways that underlie the genetic and environmental determinants of lifespan are among the most intensely studied areas in the aging field (*Kaya et al., 2015*; *Ma et al., 2018*; *de Magalhães et al., 2012*; *Li and de Magalhães, 2013*; *Ma and Gladyshev, 2017*; *van Dongen et al., 2016*; *Dato et al., 2016*). Laboratory animal models have shown that longevity can be extended by environmental (*Dillon et al., 2007*; *Johnston and Snell, 2016*; *Packer and Fuehr, 1977*; *Bisschops et al., 2015*), dietary (*Alic and Partridge, 2011*; *Stefana et al., 2017*; *Kitada et al., 2019*), pharmacological (*Bitto et al., 2016*; *Novelle et al., 2016*; *Rajman et al., 2018*; *Nadon et al., 2017*), and genetic interventions (*Zwaan, 1999*; *Vijg and Suh, 2005*; *Singh et al., 2019*). However, many of these laboratory-adapted populations are constrained by genetic and environmental background (*Harper, 2008*; *de Magalhães, 2014*; *Reichard, 2016*). For example, artificially created mutant strains may show longer lifespan under laboratory settings but demonstrate reduced fitness in their natural environment (*Bartke et al., 2001*; *Jenkins et al., 2004*; *Chen et al., 2007*; *Delaney et al., 2011*). A more integrated approach is needed to understand how the natural environment and natural selection interact to shape lifespan and associated life-history traits.

In order to better understand the impact of natural genetic variation on lifespan, we studied 76 wild isolates of the budding yeast to capture the molecular signatures of evolved diversity of lifespan. This collection included 40 diploid isolates of *Saccharomyces cerevisiae* and 36 diploid isolates of *Saccharomyces paradoxus* (*Hyma and Fay, 2013*; *Liti et al., 2009*). Their niches include human-associated environments, such as breweries and bakeries, and different types of wild ecological niches, such as trees, fruits, vineyards, and soils across different continents. There was also a group of clinical isolates; *S. cerevisiae* strains isolated from immunocompromised patients. *S. cerevisiae* and *S. paradoxus* are closely related (share a common ancestor between 0.4 and 3 million years ago), with 90% genome identity, and can mate and produce viable progeny (*Liti et al., 2006*; *Tirosh et al., 2009*). Earlier genome sequencing of these isolates revealed allelic profiles, their ploidy status, and a phylogeny of these isolates (*Hyma and Fay, 2013*; *Liti et al., 2009*; *Liti et al., 2006*). Based on these studies, it was shown that while some of these strains fall into distinct lineages with unique genetic variants, almost half of the strains have mosaic recombinant genomes arising from outcrosses between genetically distinct lineages of the same species (*Liti et al., 2009*). Because of their wide ecological, geographical, and genetic diversity, natural isolates of the budding yeast *Saccharomyces* have become an important model system for population/evolutionary genomics (*Liti, 2015*) and to study the complex genetic architecture of lifespan (*Kaya et al., 2015*; *Stumpferl et al., 2012*; *Kwan et al., 2013*; *Janssens and Veenhoff, 2016*; *Jung et al., 2018*; *Barré et al., 2020*). Therefore, these strains offer a powerful genetic pool to understand how natural genetic variation may shape lifespan variation.

Accordingly, we assayed the RLS of ~3,000 individual cells representing these isolates under two different conditions: yeast peptone dextrose (YPD, with 2 % glucose), and yeast peptone glycerol (YPG, 3 % glycerol as a respiratory carbon source), and identified up to 10-fold variation in median RLS under each condition. Although little is known about the life histories of these wild isolates they face different, niche-specific evolutionary pressures for adaptation to different stresses (*Liti et al.,*

*2009*; *Liti et al., 2006*; *Tirosh et al., 2009*; *Liti, 2015*). To understand how different genotypes arrive at different lifespan phenotypes, we further analyzed endophenotypes (gene expression, metabolite abundance) to characterize the molecular patterns associated with condition-specific lifespan variation. Following characterization of transcripts and metabolites with significant association with longevity, we identified pathways associated with median RLS across these isolates. Our data showed that the naturally arising variation in genotype can cause large differences in lifespan, which are associated with distinct patterns of gene expression and metabolite abundances. These analyses revealed connected pathways that have not been previously associated with lifespan variation in a laboratory setting. Altogether, we present the most comprehensive analysis to date of how the environment and genetic variation interact to shape aging and the associated life-history traits.

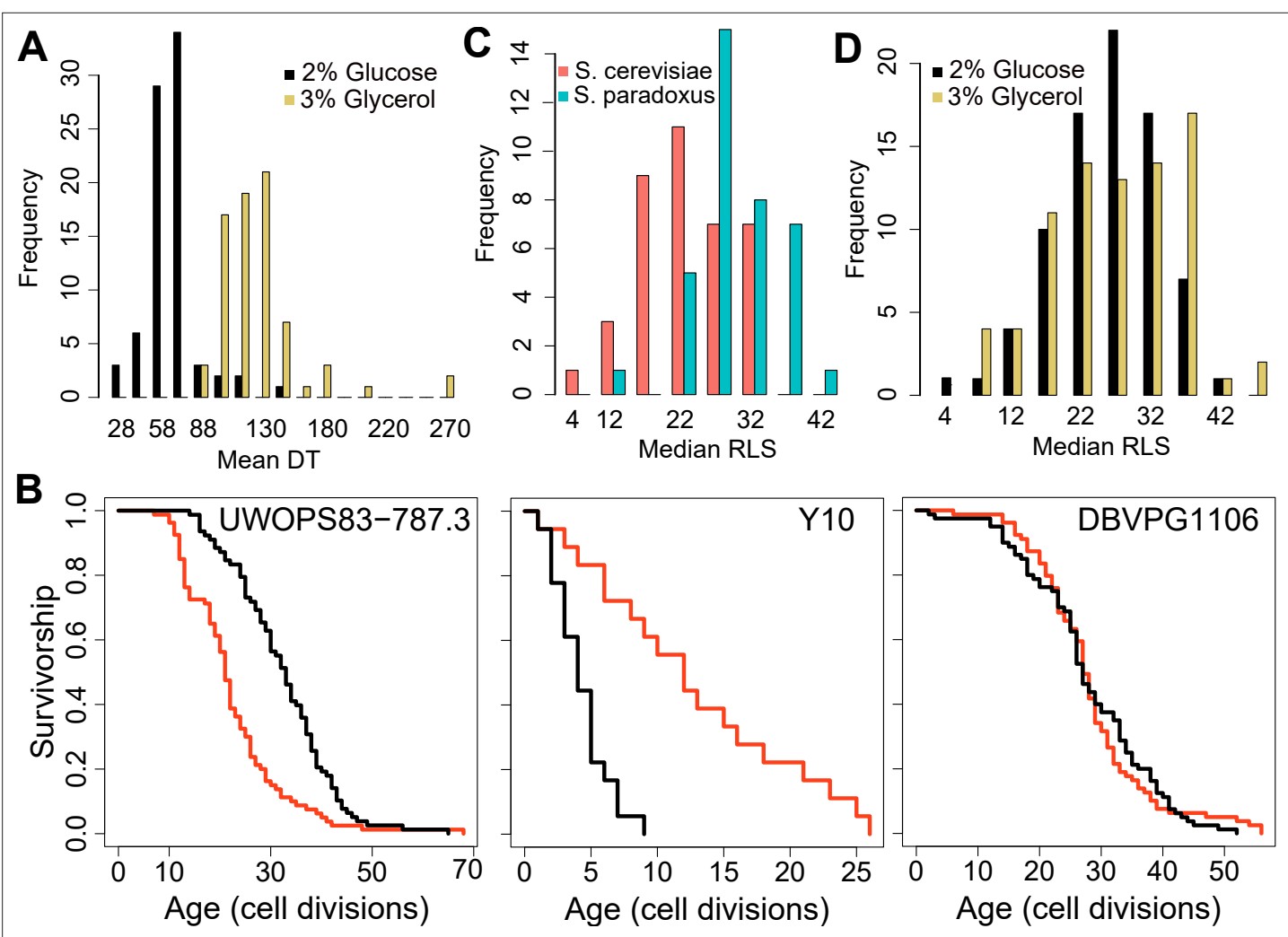

**Figure 1.** Doubling time and replicative lifespan of yeast wild isolates. (**A**) Distribution of mean doubling time (DT, in minutes) on yeast peptone dextrose (YPD; 2 % glucose) and yeast peptone glycerol (YPG; 3 % glycerol). (**B**) Examples of lifespan curves for the selected strains. Black curve shows lifespan under YPD conditions, and red curve under YPG conditions. (**C**) Median replicative lifespan (RLS) distribution across *S. cerevisiae* (red) and *S. paradoxus* (turquoise) isolates grown in YPD. (**D**) Distribution of median RLS across different conditions. Source data are provided as *Supplementary file 1*.

The online version of this article includes the following figure supplement(s) for figure 1:

**Figure supplement 1.** Replicative lifespan (RLS) phenotype across wild isolates.

## Results

### Growth characteristics and lifespan variation across wild isolates

First, we monitored growth characteristics of natural yeast isolates on standard glucose media (i.e., during fermentation) and on media with a nonfermentable substrate, glycerol, as a carbon source (i.e., during respiration), using an automated growth analyzer, and calculated the doubling time. Most wild isolates grew faster than the diploid laboratory wild-type (WT) BY4743 strain under both conditions, with an average doubling time of 65 min in YPD and 125 min in YPG (*Figure 1A*, *Supplementary file 1*). Importantly, most of these strains grew at a similar rate on YPD, and variation in growth rate on YPG was also relatively small among them, indicating that these laboratory-optimized culture conditions are suitable for supporting nutritional needs of these strains and for RLS analysis (*Figure 1A*).

Next, we assayed RLS of these isolates at 30 °C on both growth conditions (YPD and YPG media). Under the YPD condition, we observed a remarkable ~10-fold variation in median and maximum RLS (Pearson correlation coefficient = 0.65 between median and maximum RLS, p < 0.00001) across these isolates (*Figure 1B, C*, *Figure 1—figure supplement 1A*, *Supplementary file 1*). The average median RLS of *S. paradoxus* strains (29.7) was significantly higher (p = 2.7 × 10$^{-9}$) than the average median RLS of *S. cerevisiae* strains (22.7) (*Figure 1C*, *Figure 1—figure supplement 1A*). Among the 76 strains analyzed, *S. paradoxus* strain Y7 showed the longest median RLS (RLS = 42), whereas *S. cerevisiae* strain Y10 exhibited the shortest median RLS, with 50 % of Y10 cells ceasing division after producing only four daughters (*Supplementary file 1*).

Glycerol as a growth substrate can extend both RLS and chronological lifespan (CLS; *Burtner et al., 2009*; *Kaeberlein et al., 2005a*). In the case of CLS, the increased longevity is caused by a switch from fermentation to respiration *Burtner et al., 2009*; however, mechanisms by which glycerol affects RLS are unclear, since respiratory metabolism is not always required for RLS extension in laboratory WT strains (*Kaeberlein et al., 2005a*). While we observed a significant (p < 0.05, Wilcoxon rank sum test) median RLS increase in 32 strains on YPG with 24 strains significantly decreased median RLS; and the remaining 7 strains showed no significant changes (*Figure 1—figure supplement 1B*, *Supplementary file 1*). For example, *S. cerevisiae* strain BC187 showed a significantly increased median RLS on glycerol (37.5 in YPG versus 32 in YPD), and *S. paradoxus* strain KPN3829 also showed a similar increase (21 in YPD and 35 in YPG). On the other hand, *S. cerevisiae* strain YJM975 showed a significant decrease (median RLS = 31 in YPD and 23 in YPG) (*Supplementary file 1*).

We further dissected the effect of carbon source on RLS by comparing median RLS variation between YPD and YPG conditions. Interestingly, strains with the shortest RLS on YPD tended to achieve the longest lifespans on YPG, while the long-lived strains on YPD generally did not show a further RLS increase (*Figure 1—figure supplement 1B, C*) (Pearson correlation coefficient = −0.51 between median RLS on YPD and median RLS on YPG, p < 0.0001). A similar observation was shown under CR conditions in the case of single-gene deletions, where the shortest-lived strains tended to yield the largest lifespan extension when subjected to CR (*Schleit et al., 2013*). Overall, we observed significant differences in lifespan across these strains on both conditions. The observed lifespan pattern on YPG conditions in comparison to those on YPD suggests that long-lived strains might reside in environments with low fermentable carbon sources, so that they undergo distinct metabolic regulation to metabolize respiratory carbon sources. As such, they did not show further RLS extension when grown in glycerol.

### Endophenotype variation across wild isolates

While many previous large-scale omics studies on aging have focused on genome-wide association (*Stumpferl et al., 2012*; *Burke et al., 2014*; *Wilson et al., 2020*; *Deelen et al., 2019*), recent comparative studies on transcriptomics (*Fushan et al., 2015*; *Fuentealba et al., 2021*), proteomics (*Yang et al., 2008*; *Tanaka et al., 2018*; *Heinze et al., 2018*), metabolomics (*Laye et al., 2015*; *Ma et al., 2015b*; *Cheng et al., 2015*), and ionomics *Ma et al., 2015a* have begun to shed light on molecular patterns and mechanisms associated with the aging process. For example, it has been suggested that natural variation is associated with extensive changes in gene expression, translation, and metabolic regulation, which in turn may affect fitness under different stress conditions (*Whitehead, 2006*; *Vu et al., 2015*). In fact, gene expression variation has repeatedly been postulated to play a major role in adaptive evolution and phenotypic plasticity (*Vu et al., 2015*; *Gilad et al., 2006*), as well as specific phenotypic outcomes such as changes in morphology (*Beldade et al., 2002*) and lifespan (*Whitaker*

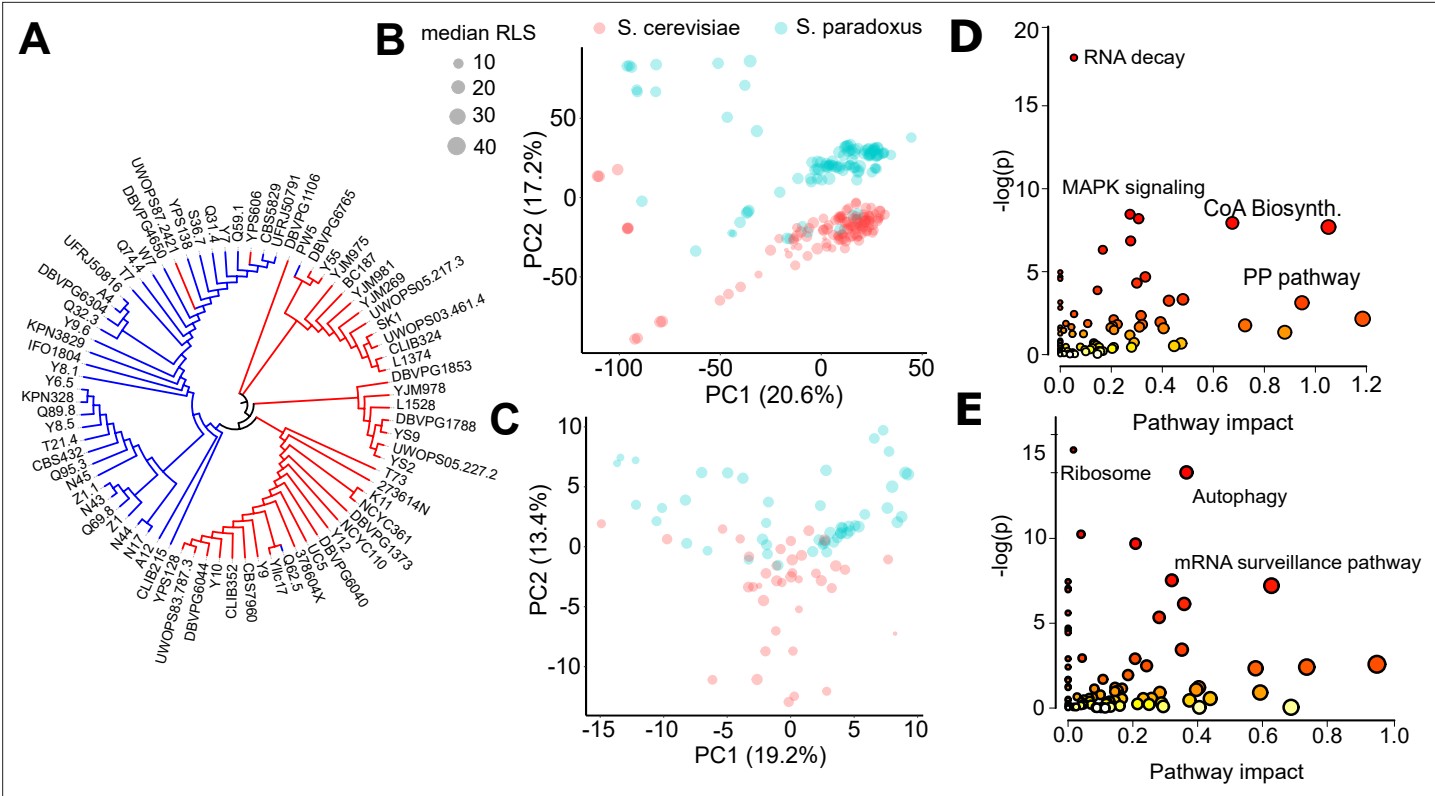

**Figure 2.** Endophenotypic variation across strains. (**A**) Phylogenetic relationship based on the transcriptome data of 76 strains of 2 species. Principal component analysis (PCA) of (**B**) transcriptomics and (**C**) metabolomics. Percent variance explained by each principal component (PC) is shown in parentheses. Pathway enrichment analysis for combined top genes and metabolites contributing to (**D**) PC1 and (**E**) PC2. Some of the enriched Kyoto Encyclopedia of Genes and Genomes (KEGG) pathways are shown in each panel. Source data are provided as *Supplementary file 3*.

The online version of this article includes the following figure supplement(s) for figure 2:

**Figure supplement 1.** Read alignment rate for RNA-seq data.

**Figure supplement 2.** Correlation of genome-wide transcript levels across wild isolates.

**Figure supplement 3.** Principal component analysis.

*et al., 2014*). Similarly, comparative studies of metabolite profiles have been utilized to describe the genotype to phenotype relations in model organisms (*Guijas et al., 2018*; *Harrison et al., 2020*). Accordingly, we aimed to explain lifespan differences among these wild-derived yeast isolates by analyzing their gene expression variation (based on transcriptomics analyses) and differences in their metabolite levels (based on metabolomics analyses).

In the case of the transcriptome, we obtained ≥5 million 150 bp paired-end RNA-seq reads for each strain grown on YPD. For metabolomics analyses, we applied targeted metabolite profiling using liquid chromatography–mass spectrometry (LC–MS). After filtering and quality control, the dataset contained RNA-seq reads for 5376 genes and 166 metabolites identified commonly across all isolates (*Supplementary file 2*). The expression profiles between strains were similar to one another, with Spearman correlation coefficients of strain pairs ranging between 0.59 and 0.93 (except the pairing involving Q59.1, CBS5829, YPS606, and UFRJ50791, with the range between 0.21 and 0.79) (*Figure 2—figure supplements 1 and 2*). To determine whether the previously published sequence-based evolutionary relationships (*Liti et al., 2009*) were reflected in their gene expression variation, we constructed gene expression phylograms using a distance matrix of 1 minus Spearman correlation coefficients and the neighbor-joining method (*Brawand et al., 2011*). The resulting topology of species-specific trends was largely consistent with their phylogeny with a clear separation between *S. cerevisiae* and *S. paradoxus* strains (*Figure 2A*); however, at the intraspecies level, the topology was not consistent with the phylograms of genomic data.

To visualize endophenotypic variation between these two species and across the strains of the same species, we performed principal component analysis (PCA) on each type of data. PCA of the transcriptome revealed a pattern resembling the phylogenetic relationship, with the first three principal components (PCs explaining ~49 % of total variance in gene expression (*Figure 2—figure supplement 3A*)). Although some *S. paradoxus* strains clustered with *S. cerevisiae*, we observed clear species segregation based on PC2 (except for some outlier strains that were separated by PC1). PCA of metabolomics data revealed a similar structure with the first three PCs explaining ~41 % of total variance in metabolite levels, and PC2 somewhat separating the species (*Figure 2—figure supplement 3B*).

To understand the basis of this segregation pattern, we performed pathway enrichment analysis by combining the 500 top genes (250 with positive weights and 250 with negative weights) and 40 top metabolites (20 with positive weights and 20 with negative weights) contributing to each PC, respectively. This integrative analysis of genes and metabolites (see Materials and methods) contributing to PC1 revealed a distinct set of Kyoto Encyclopedia of Genes and Genomes (KEGG) pathways, including RNA degradation, mitogen-activated protein kinase (MAPK) signaling pathway, cell cycle, pantothenate and CoA biosynthesis, ribosome biogenesis, and pentose phosphate pathway (*Figure 2D*, *Supplementary file 3*). The analysis of genes and metabolites for PC2 revealed the KEGG pathways related to ribosome, autophagy, endocytosis, cell cycle, mRNA surveillance, and nucleotide excision repair (*Figure 2E*, *Supplementary file 3*). These results suggest that these processes diverged most significantly across the wild isolates of two species of *Saccharomyces* genus and may account for their phenotypic diversity, including lifespan. It should be noted, however, that we do not know exactly what each PC represents, unless it perfectly aligns or correlates with some known variables. In

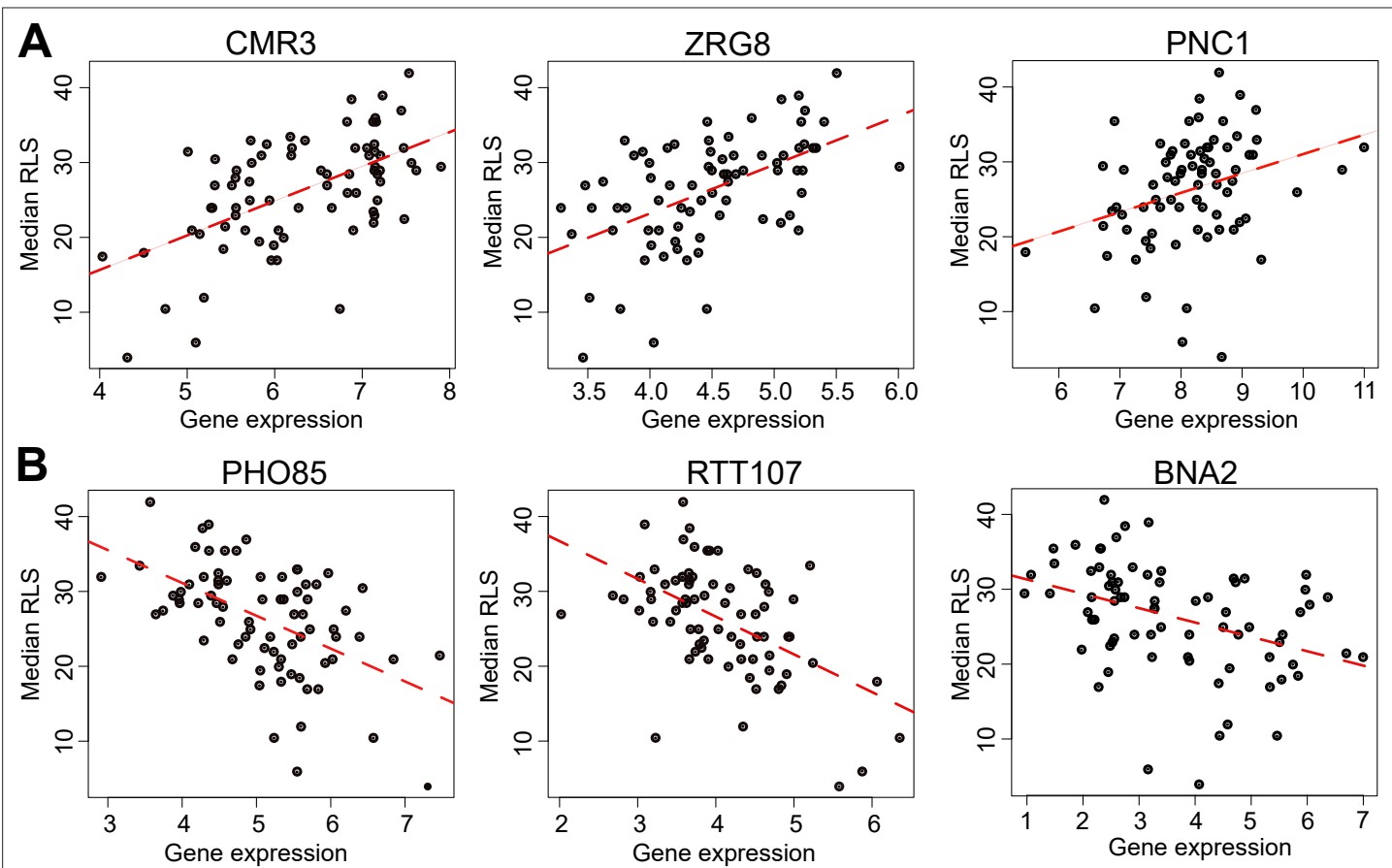

**Figure 3.** Selected genes whose expression correlates with median replicative lifespan (RLS). (**A**) Gene expression level (log2-cpm) of *CMR3, ZRG8*, and *PNC1* positively correlates with median RLS. (**B**) Transcript abundance of *PHO85, RTT107*, and *BNA2* negatively correlates with median RLS. Regression slope p values can be found in *Supplementary file 2*, which is also the source data file for these analyses.

addition, either biological (e.g., phylogenetic structure), technical (e.g., data normalization or batch effect), or mixed effects of both may render PCA biased (*Uyeda et al., 2015*).

## Relationship between endophenotypes and lifespan

To identify endophenotypes (transcripts and metabolites) correlating with lifespan variation across wild isolates, we applied the phylogenetic generalized least squares (PGLS) method to account for phylogenetic relationships among the strains and test for different models of trait evolution (*Felsenstein, 1985*; *Freckleton et al., 2002*). Regression was performed between endophenotypic values and median RLS under different models of trait evolution and the best-fit model was then selected based on maximal likelihood. To assess the robustness of these relationships, we repeated the regression after taking out one yeast strain at a time and only those regressions that remained significant were further considered. This ensured the overall relationship did not depend on a particular isolate.

With the PGLS approach, we identified 73 transcripts with significant correlation with median RLS ($p_{adj}$ ≤ 0.01; 39 with positive correlation and 34 with negative correlation) (*Supplementary file 2*). Among the top hits with positive correlation were a putative zinc finger protein coding gene *CMR3* ($p_{adj} = 3.3 \times 10^{-9}$), histone acetyltransferase (HAT) gene *HPA2* ($p_{adj} = 0.0002$), transcription factor *TEC1* ($p_{adj} = 9.3 \times 10^{-6}$), and zing regulated protein gene *ZRG8* ($p_{adj} = 0.006$) (*Figure 3A*, *Supplementary file 2*). The top hits with negative correlation included the genes coding for cyclin-dependent kinase Pho85p interacting proteins *PCL1* ($p_{adj} = 0.0008$) and *PCL2* ($p_{adj} = 0.001$), regulator of Ty1 transposon protein coding gene *RTT107* ($p_{adj} = 0.007$), and inositol monophosphatase gene *INM1* ($p_{adj} = 0.006$) (*Figure 3B*, *Supplementary file 2*). Next, to assess if any of our transcript hits were previously implicated in yeast lifespan, we extended our list of significant genes to 357 genes by selecting a cutoff at $p_{adj} = 0.05$ and compared these with the genes associated with RLS in laboratory WT strain listed in the GenAge database (*de Magalhaes, 2009*). GenAge identifies 611 genes from the published literature with effects on RLS (decreased or increased) of laboratory yeast strains (595 deletion mutants and 16 overexpressed genes) (*Supplementary file 4*). Of 5376 genes whose expression was measured across the wild isolates, there were 39 genes present in both our list and GenAge, 23 of which showed the same direction of correlation with RLS. For example, *INM1*, *RTT107*, *PPH3*, and *BSC1* genes increase RLS when deleted GenAge database (*Supplementary file 4*) and are associated with increased RLS when their transcript levels decrease across wild isolates (this study). However, the overall pattern of overlapping genes as well as the direction of correlation did not reach statistical significance (Fisher's exact test, $p \geq 0.05$). It should be noted that many of the RLS-associated genes listed in GeneAge are reported from single-gene knockout (KO) studies and there has been no comprehensive studies examining gene overexpression on a genome-wide scale. This raises a possibility that the genes we identified here might not necessarily be overrepresented among the lifespan-related genes from other studies. It is also possible that the genetic architecture of trait variation in natural populations may differ from that which is assumed from studies of lab strains, including extensive single-gene studies of lifespan variation in yeast (*McCormick et al., 2015*). The lack of overlap between the genes whose expression correlates with lifespan variation in wild isolates and genes that affect RLS in single-mutant studies on laboratory WT background supports this possibility. Considering this, we then asked if trait variation in wild isolates and lab strains may converge at the transcriptome in a way that may be detectable at the level of gene expression, or at the level of biological pathway. To do this we examined the gene expression patterns across wild isolates with those of 1376 laboratory KO strains *Kemmeren et al., 2014* whose RLS was quantified (*McCormick et al., 2015*) previously (*Figure 4—figure supplement 1*, *Supplementary file 1*). We calculated an association of gene expression with different measures of RLS (mean RLS, median RLS, and maximum RLS) across KO strains. Our analysis revealed around 400 significant genes ($p_{adj} = 0.05$) associated with three types of RLS measures, and more than 1000 genes associated with median RLS (*Figure 4A*, *Supplementary file 2*). To compare the RLS-associated transcriptomes of wild isolates and lab strains, we then calculated a correlation matrix of RLS-associated gene expression changes across KO strains and wild isolates (*Figure 4A*). We found no positive correlation between RLS-associated gene expression changes across KO strains and RLS-associated gene expression changes across natural isolates (*Figure 4B*).

We then performed functional enrichment (gene set enrichment analysis, GSEA) of genes associated with RLS across deletion and wild isolates to see if associations with RLS may converge at the level of the biological pathway. We find that the transcripts associated with RLS in these two

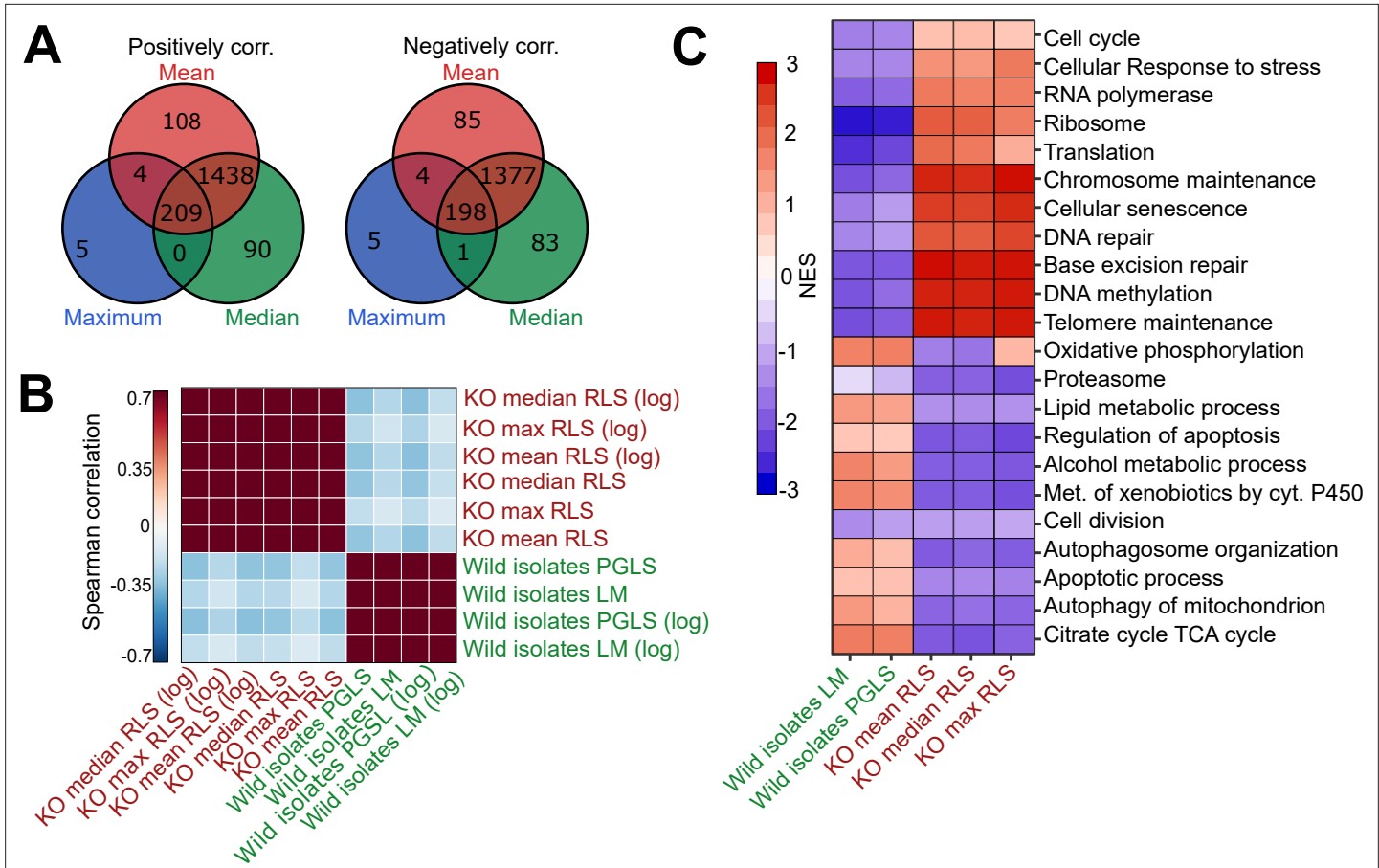

**Figure 4.** Comparative analysis of lab yeast knockout (KO) and wild isolates. (**A**) Significant genes ($p_{adj}$ < 0.05) associated with maximum, median, and mean replicative lifespan (RLS; or log maximum, median, and mean) across deletion strains based on transcriptomics data obtained from 1376 KO strains. Genes positively and negatively associated with RLS (upregulated and downregulated, respectively) are significantly shared across different metrics of RLS (Fisher's exact test, p < 0.05). (**B**) Denoised correlation matrix of gene expression effects across single-gene deletion strains (KO), and those that we measure across the wild isolates that are associated with RLS. Correlation coefficient is calculated using union of top 1000 statistically significant genes for each pair of signatures with Spearman method. LM: linear model; PGLS: phylogenetic regression least squares. (**C**) Functional enrichment of genes associated with RLS across deletion and natural strains. Cells are colored based on normalized enrichment score (NES). *Supplementary files 1 and 2* are provided as source data files for these analyses.

The online version of this article includes the following figure supplement(s) for figure 4:

**Figure supplement 1.** Replicative lifespan (RLS) phenotype of yeast knockout strains.

populations enrich distinct sets of biological pathways (*Figure 4C, Supplementary file 2*). For the genes correlating positively with longevity across the KO strains, the enriched terms included cellular responses to stress, ribosome, translation, cellular senescence, and DNA repair (*Figure 4C, Supplementary file 2*). On the other hand, terms enriched in wild isolates included TCA cycle, oxidative phosphorylation, and lipid metabolic process, regulation of apoptosis, and autophagy (*Figure 4C, Supplementary file 2*). Overall, our comparative analyses of lifespan-associated gene expression signatures in laboratory-adapted yeast strains versus wild isolates suggest that different genetic trajectories might have evolved at transcript level across wild isolates to regulate lifespan.

Next, we searched for metabolites whose abundances associate with RLS across wild yeast isolates. The metabolome represents a snapshot of regulation downstream of both the transcriptome and proteome, and it has been effectively used for characterizing phenotypic variation that includes lifespan (*Laye et al., 2015; Ma et al., 2015b; Cheng et al., 2015*). Among 166 metabolites that we examined, 31 exhibited significant association with median RLS ($p_{adj}$ ≤ 0.05) (*Supplementary file 2*). Among the top hits, tryptophan, lactate, 2-hydroxyglutarate, 3-hydroxypropionic acid, 2-hydroxyisobutyrate, 2-hydroxybutyrate, and phenyl-lactic acid (PLA) correlated positively (*Figure 5A*), whereas lysine,

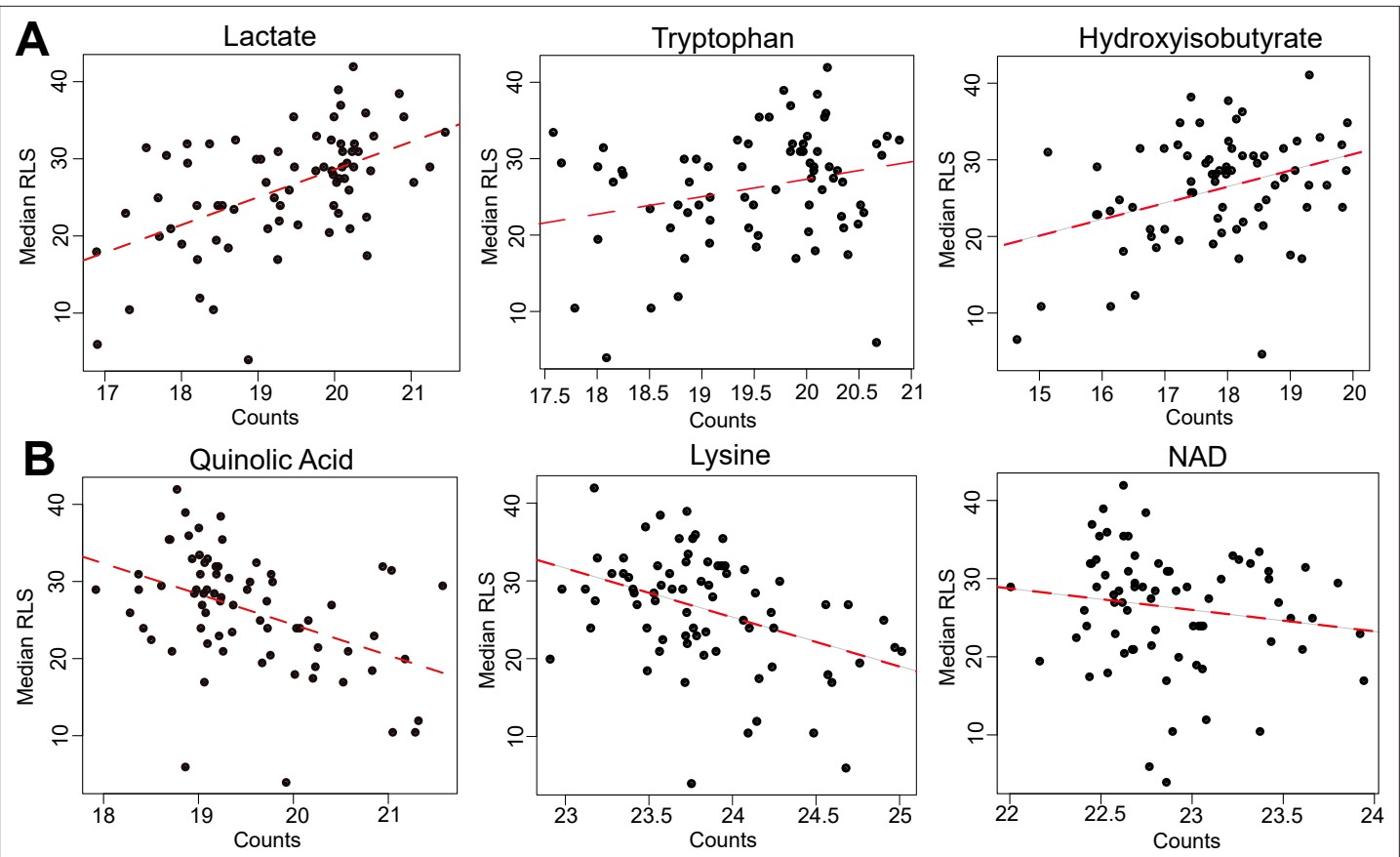

**Figure 5.** Selected metabolites correlating with median replicative lifespan (RLS). (**A**) Abundance (liquid chromatography–mass spectrometry [LC–MS] counts) of lactate, tryptophan (Trp), and hydroxyisobutyrate that positively correlate, and (**B**) abundance of quinolinic acid, lysine (Lys), and NAD that negatively correlate with median RLS. Regression slope p values can be found in ***Supplementary file 2***. This file is also provided as a source data file for these analyses.

The online version of this article includes the following figure supplement(s) for figure 5:

**Figure supplement 1.** Correlation of LYS20 and LYS21 genes with median replicative lifespan (RLS).

quinolinic acid (QA), propionate, Se-methylselenocysteine showed negative correlation (***Figure 5—figure supplement 1***). Our metabolite list also included several related short-chain fatty acids, with positive correlation to RLS short cahin fatty acids (SCFAs: 3-hydroxypropionic acid, 2-hydroxyisobutyrate, 2-hydroxybutyrate, and 2-hydroxyglutarate), which are known to be involved in redox regulations, epigenetic modification, and energy generation (***He et al., 2020***; ***Tan et al., 2014***). Having identified transcripts and metabolites associated with lifespan, we aimed to investigate interaction among them to better understand biological causes of lifespan variation and mechanisms of longevity across the wild isolates.

## Molecular signatures of lifespan extension across wild isolates

To understand the molecular basis of RLS variation in wild isolates, we applied an integrated pathway approach (***Pang et al., 2021***). A combined metabolomics and transcriptomics data analysis revealed a potential role for differential metabolic regulation of tryptophan (Trp), lysine (Lys), and branched chain amino acid (BCAAs) biosynthesis as well as valine (Val) and isoleucine (Iso) (***Figure 6—figure supplements 1–3***). For example, our metabolome data revealed that Trp abundance correlates positively with RLS (***Figure 5A***). In addition, we found that QA, an intermediate in the Trp catabolic pathway (also known as the kynurenine [KYN] pathway) (***Platten et al., 2019***; ***Pinson et al., 2019***) correlates negatively with median RLS (***Figure 5B***), suggesting a possible inhibition of Trp degradation, corresponding to an increased Trp abundance in long-lived strains. Further evidence for metabolic regulation of Trp in long-lived strains came from our transcriptome data wherein *BNA2* (indoleamine

2,3-dioxygenase) gene, which supports the first rate-limiting step of Trp catabolism (*Pinson et al., 2019*), was significantly downregulated ($p_{adj}$ = 0.02) in long-lived wild isolates (*Figure 3B*). These observations draw a complete picture for the observed Trp abundance in long-lived strains.

Similarly, a link between our transcriptome and metabolome data provided insights into Lys metabolism. We observed a negative correlation between Lys abundance and lifespan (i.e., long-lived strains tend to have less Lys) (*Figure 5B*). Additionally, our transcriptome data showed negative correlations of two homocitrate synthase genes, *LYS20* and *LYS21*, controlling the first rate-limiting step of Lys biosynthesis by catalyzing condensation of Acetyl-CoA and alpha-ketoglutarate (α-KG) to produce homocitrate (*Figure 5—figure supplement 1*). Together, these observations support the idea of decreased Lys levels in long-lived strains. It is also of interest that while they did not reach significance, all genes (with the exception of *ARO8*) involved in Lys biosynthesis showed a trend for decreased expression in long-lived strains (*Supplementary file 2*). Interestingly, previous studies have shown the connection between Trp and Lys metabolism both at genetic and metabolic levels. For example, it has been shown that 3-hydroxyanthranilic acid, an intermediate from Trp degradation, can be used as a substrate to synthesize α-ketoadipate (*Hallen et al., 2013*; *Tobes and Mason, 1975*), which is then converted by *ARO8* to Lys (*Figure 6—figure supplements 1 and 2*). In this regard, glutamate (Glu)-dependent *ARO8* activity is involved in both Trp and Lys catabolic pathways. In addition, at the genetic level, while individual KO lines of *BNA2*, *LYS20*, or *LYS21* are not lethal, it was found that the combined deletion of *BNA2* and *LYS20* or *BNA2* with *LYS21* causes synthetic lethality (*Deutscher et al., 2006*), possibly by causing Lys auxotrophy. In the light of these observations, our data suggest that the observed occurred differences in Trp and Lys metabolism are not random. Decreased Trp catabolism in long-lived strains might limit α-ketoadipate production, which in turn could affect Lys biosynthesis.

Our analyses also revealed negative correlations between RLS and transcript abundance for all genes (negative correlation) involved in BCAA biosynthesis from pyruvate in long-lived strains (*Figure 6—figure supplement 3*, *Supplementary file 2*). On the other hand, we did not observe any changes in Val, Leu, and Ile abundance. This observation raises a possibility that intracellular homeostasis of these BCAAs might be regulated through other resources (e.g., extracellular import) (*Hammer and Avalos, 2017*). The other metabolites that showed a significant correlation to RLS were lactic acid (LA), PLA, tyrosine (Tyr), and aspartic acid (Asp) (*Figure 6—figure supplement 1*, *Supplementary file 2*). Although the synthesis of LA from pyruvate is well studied, the metabolic regulation and function of PLA, a product of the shikimate pathway, are less clear. Previously, it was found that yeast produces PLA through a nonspecific activity of lactate dehydrogenase from phenylpyruvate, a metabolite derived from chorismate in the shikimate pathway (*Srinivasan and Smolke, 2020*). Interestingly, Tyr is also synthesized via the shikimate pathway (*Figure 6—figure supplement 1*) and its abundance negatively correlates with RLS. The decreased *ARO2* (synthesizes chorismate from shikimate) expression might explain the decreased abundance of Tyr in long-lived strains. Similarly, one can expect a decreased Trp abundance, which is also synthesized through the shikimate pathway. However, our data revealed an increased level of Trp in long-lived strains. Therefore, we relate this observation to the decreased *BNA2* expression (see above).

Finally, since amino acid metabolism is directly related to glycolytic and/or TCA cycle intermediates (*Figure 6—figure supplements 1–3*, *Supplementary file 2*), we analyzed differences in metabolites and genes involved in these central metabolic processes between short- and long-lived strains. Analysis of the data based on metabolomics and transcriptomics approaches suggested a decreased glycolytic rate and increased TCA cycle activity in long-lived strains (*Figures 4C and 6A*). For example, we found that glycolytic genes such as *FBA1*, *TDH2*, *PGK1*, *ENO2*, and *CDC19* were negatively correlated with RLS, while TCA cycle genes such as *CIT1*, *IDP1*, and *KGD1* were positively correlated with RLS (*Supplementary file 2*). These observations are consistent with the pathway enrichment analysis revealing increased TCA cycle and oxidative phosphorylation in long-lived strains. To further examine this, we measured basal oxygen consumption rate (OCR) of wild yeast isolates and verified the increased respiration rate in long-lived strains. To determine if the observed pattern of median RLS variation can be partly explained by this increased mitochondrial function, we tested a potential relationship between OCR and median RLS. Our analysis revealed a significant positive correlation (*R* = 0.28, $p_{adj}$ = 0.016) between OCR and median RLS (*Figure 6—figure supplement 4A*, *Supplementary file 5*). Furthermore, we tested whether the increased OCR can be simply explained by total

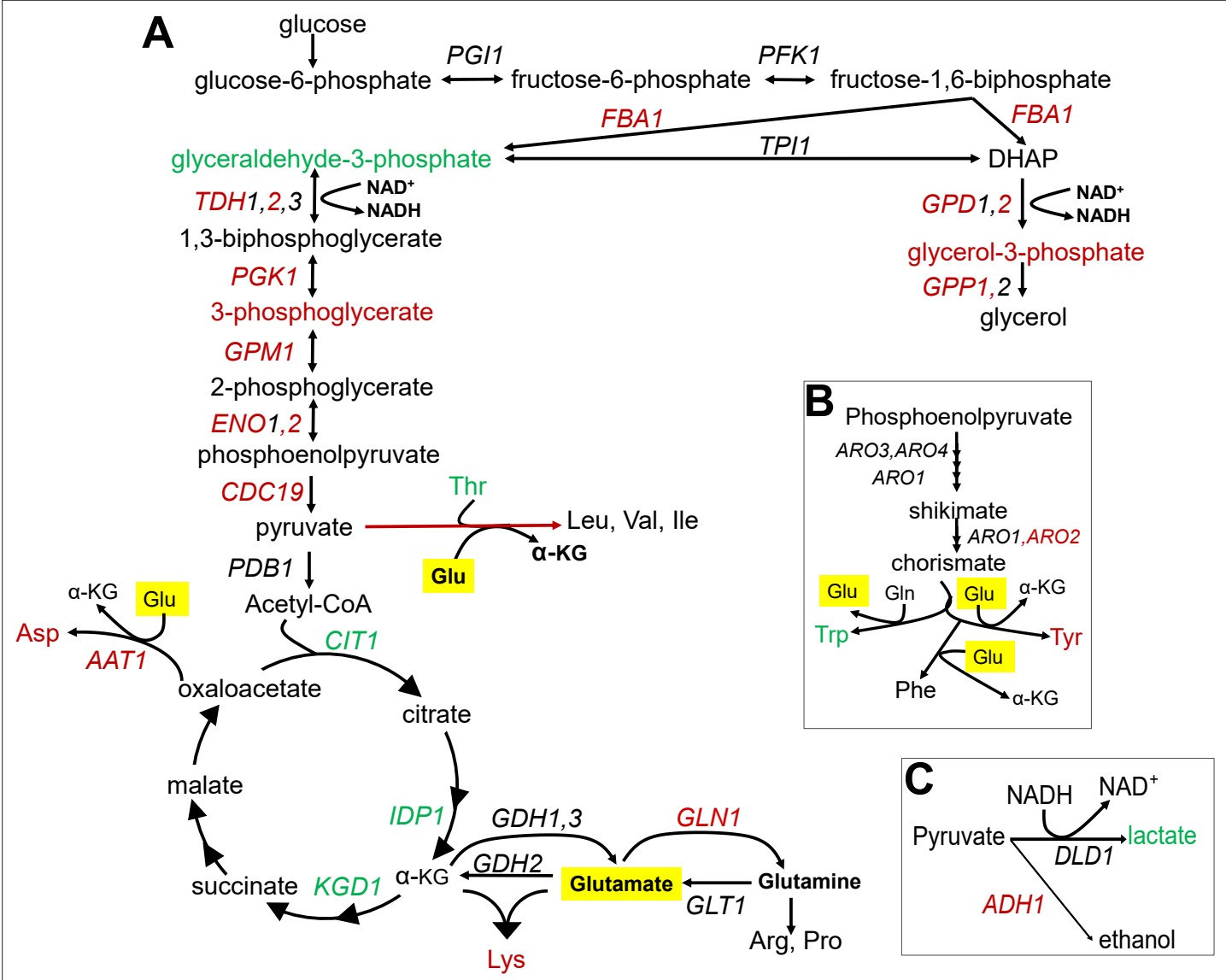

**Figure 6.** Summary of metabolic changes associated with replicative lifespan (RLS). (**A**) Summary depiction of genes and metabolites from the interconnected glycolytic pathway, TCA cycle and amino acid metabolism that are found to be associated with RLS. Associated genes are colored in red (negatively associated with RLS) or green (positively associated with RLS). Depiction of (**B**) shikimate pathway and (**C**) lactate and ethanol biosynthetic pathways are shown with the same color code representation. Glutamate is highlighted in yellow.

The online version of this article includes the following source data and figure supplement(s) for figure 6:

**Figure supplement 1.** Genes and metabolites from shikimate, kynurenine, and salvage pathways associated with replicative lifespan (RLS).

**Figure supplement 2.** Lysine biosynthesis and replicative lifespan (RLS).

**Figure supplement 3.** Genes from the branched chain amino acid (BCAA) metabolic pathway are negatively associated with replicative lifespan (RLS).

**Figure supplement 4.** Association of mitochondrial respiration with median replicative lifespan (RLS).

**Figure supplement 4—source data 1.** Raw western Blot images are provided as source data.

mitochondrial copy number by analyzing protein abundance of mitochondrial marker protein Por1 by western blots. We observed a similar abundance of Por1 across the strains, arguing against alteration in mitochondrial copy number in long-lived strains (*Figure 6—figure supplement 4—source data 1*). Overall, these data suggest a possible role of mitochondrial function in lifespan variation across wild yeast isolates.

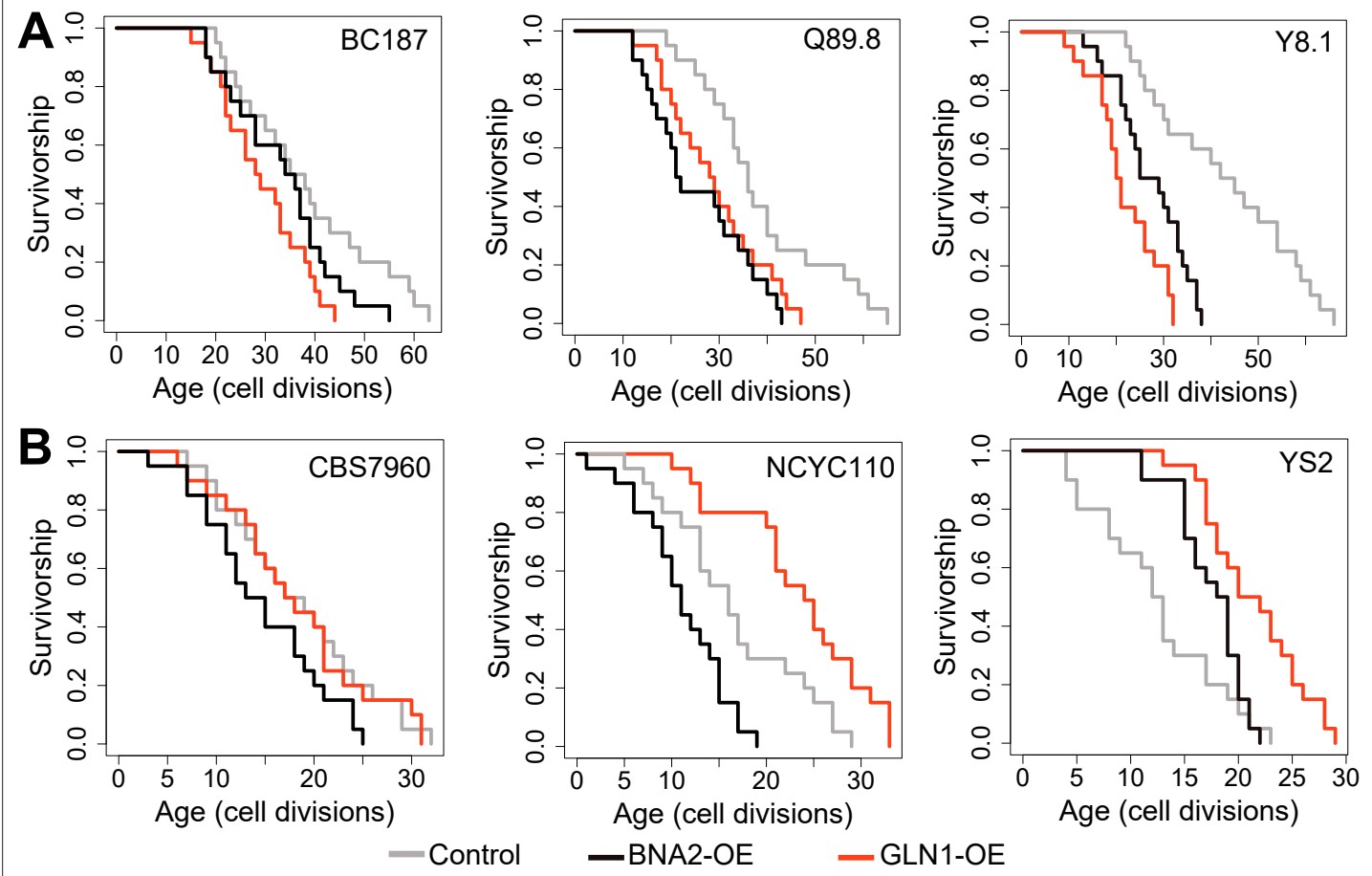

**Figure 7.** Replicative lifespan (RLS) effect of *GLN1* and *BNA2* overexpression in selected long- and short-lived strains. Lifespan curves for control (gray), *BNA2* (black), and *GLN1* (red) overexpression in (**A**) long- and (**B**) short-lived strains. Lifespan data and significance of lifespan changes can be found in ***Supplementary file 1***.

In summary, our joint omics analyses revealed consistent changes associated with increased lifespan at both metabolome and transcriptome levels, pointing to decreased glycolytic activity and amino biosynthesis and increased mitochondrial activity (TCA cycle and mitochondrial respiration) even under conditions of excess fermentative carbon source (glucose). These findings suggest common changes responsible for modulating lifespan across a broad diversity of wild yeast isolates.

## Experimental testing of Glu and Trp metabolism in regulation of longevity

To further understand the molecular mechanisms that support the long life of yeast wild cells, we paid particular attention to the association between decreased Trp degradation (KYN pathway) and RLS. Our data highlight the importance of Trp metabolism in lifespan regulation in long-lived strains. We found that even though the Trp biosynthesis pathway (shikimate pathway) is suppressed, Trp levels were increased, possibly due to decreased transcript abundance of *BNA2*, which controls the first rate-limiting step in Trp degradation. These data suggest that Trp abundance itself might be important for longevity and that long-lived strains might compensate for decreased Trp biosynthesis by inhibiting Trp degradation. To test this idea, we examined the lifespan effect of increased *BNA2* dosage in three long- and short-lived strains. Consistent with the findings from transcriptomic data, the increased expression of *BNA2* caused a significant decrease in median RLS in two out of three long-lived strains tested and significantly decreased maximum RLS in all long-lived strains tested (***Figure 7A***, ***Supplementary file 1***). The increased expression of *BNA2* in short-lived strains did not result in a consistent RLS pattern. Among the three short-lived strains tested, two decreased median and maximum RLS

significantly, and caused a significant increase in median RLS in one strain (*Figure 7B*, *Supplementary file 1*). Additional data are needed to fully clarify whether changes in Trp levels or some intermediate metabolites from KYN pathway such as QA are critical for the observed lifespan variation; however, our data support a role for Trp homeostasis in longevity.

The observation of decreased glycolysis, increased SCFAs abundance, and increased LA synthesis from pyruvate in long-lived strains seems to disagree with the findings from transcriptome and the experimental work that suggest increased TCA cycle activity and respiration. While our findings from both metabolomics and transcriptomics data suggest a decreased substrate availability for the TCA cycle, increased TCA cycle activity may be fueled by alternative substrates. We hypothesize that compartment-specific glutamate (Glu) to α-KG flux, a reaction mainly controlled by $NAD^+$-dependent mitochondrial Glu dehydrogenase, *GDH2* in mitochondria (*Han et al., 2019*; *Sickmann et al., 2003*; *Mara et al., 2018*), might support increased TCA activity. In this case, spared Glu (due to decreased Glu-dependent amino acid biosynthesis) can support citrate synthesis to fuel TCA cycle via α-KG conversion. Along with decreased glycolysis and a decrease in Glu-dependent amino acid biosynthesis, compartment-specific Glu to α-KG flux might be important for extended longevity in long-lived strains. In fact, our findings suggest that Glu utilization is limited in long-lived strains; however, the observation of no significant alteration in Glu abundance is consistent with the idea that long-lived strains may utilize Glu in some other pathway. In support of this model, we found that transcript abundance of *GLN1*, an enzyme responsible for synthesis of glutamine (Gln) from Glu (*Figure 6*) in mitochondria, negatively correlates with lifespan (*Supplementary file 2*). To test the possibility that *GDH*-mediated Glu to α-KG flux is important for supporting the lifespan of long-lived strains, we overexpressed *GLN1* in three long- and three short-lived strains. Overexpression of *GLN1* is expected to decrease the Glu pool, and thus perhaps α-KG synthesis. We found that *GLN1* overexpression significantly decreased both median and maximum RLS of long-lived strains tested (Wilcoxon rank sum tests, $p < 0.05$) (*Figure 7A*, *Supplementary file 1*). Among the three short-lived strains tested, two significantly increased median and maximum RLS, while the remaining strain showed no significant lifespan changes (*Figure 7B*, *Supplementary file 1*). Thus, the data support the idea that the mitochondrial Glu pool may have a role in longevity across wild isolates. Although it needs additional experimental evidence, we think that increased nicotinamide adenine dinucleotide (NAD) + hydrogen (H) (NADH) levels in long-lived strains might be due to increased $NAD^+$-dependent *GDH2* activity, which catalyzes the conversion of Glu to α-KG in mitochondria.

Although initially yeast was considered as Krebs negative (i.e., cannot utilize TCA cycle intermediates as carbon sources for growth) (*Casal et al., 2008*), later on α-KG was shown to be catabolized, under the condition of co-consumption with low glucose (*Zhang et al., 2020*). To investigate whether wild isolates can utilize α-KG as an alternative carbon source, we cultured them in the medium containing low glucose and α-KG, α-KG only, and YP (yeast extract peptone without glucose) medium (*Figure 8—figure supplement 1*). To our surprise, we found that many of these wild isolates showed weak growth even on the YP medium, which was not supplemented with any carbon source (*Figure 8—figure supplement 1*). It is possible that some compounds in yeast extract may promote weak growth of these isolates, and we think that it might be α-KG. To prove this, we supplemented YP and YPD medium with α-KG (10 g/l) and observed that many of the strains showed improved growth, further supporting utilization of α-KG for growth on the medium lacking glucose (*Figure 8—figure supplement 1*).

Finally, to connect increased respiration, α-KG utilization and extended lifespan, we eliminated mitochondrial DNA (mtDNA, $rho^0$) in three long-lived strains. We assayed their growth in medium supplanted with α-KG. We found that elimination of mtDNA in long-lived strains abolished their growth ability in the medium supplemented with α-KG as a sole carbon source (*Figure 8A*). We further measured RLS of these $rho^0$ isolates under 2 % glucose conditions to understand whether blocking respiration would affect their lifespan. Our analysis revealed that the loss of mtDNA caused a significant reduction in RLS in all three strains tested (Wilcoxon rank sum tests, $p < 0.05$) (*Figure 8B*, *Supplementary file 1*). Overall, these data further support the idea that α-KG utilization and increased mitochondrial respiration are connected to each other and utilization of α-KG requires active mitochondria. Perhaps, under the conditions of decreased amino acid synthesis α-KG utilization could increase respiration, which in turn may increase lifespan.

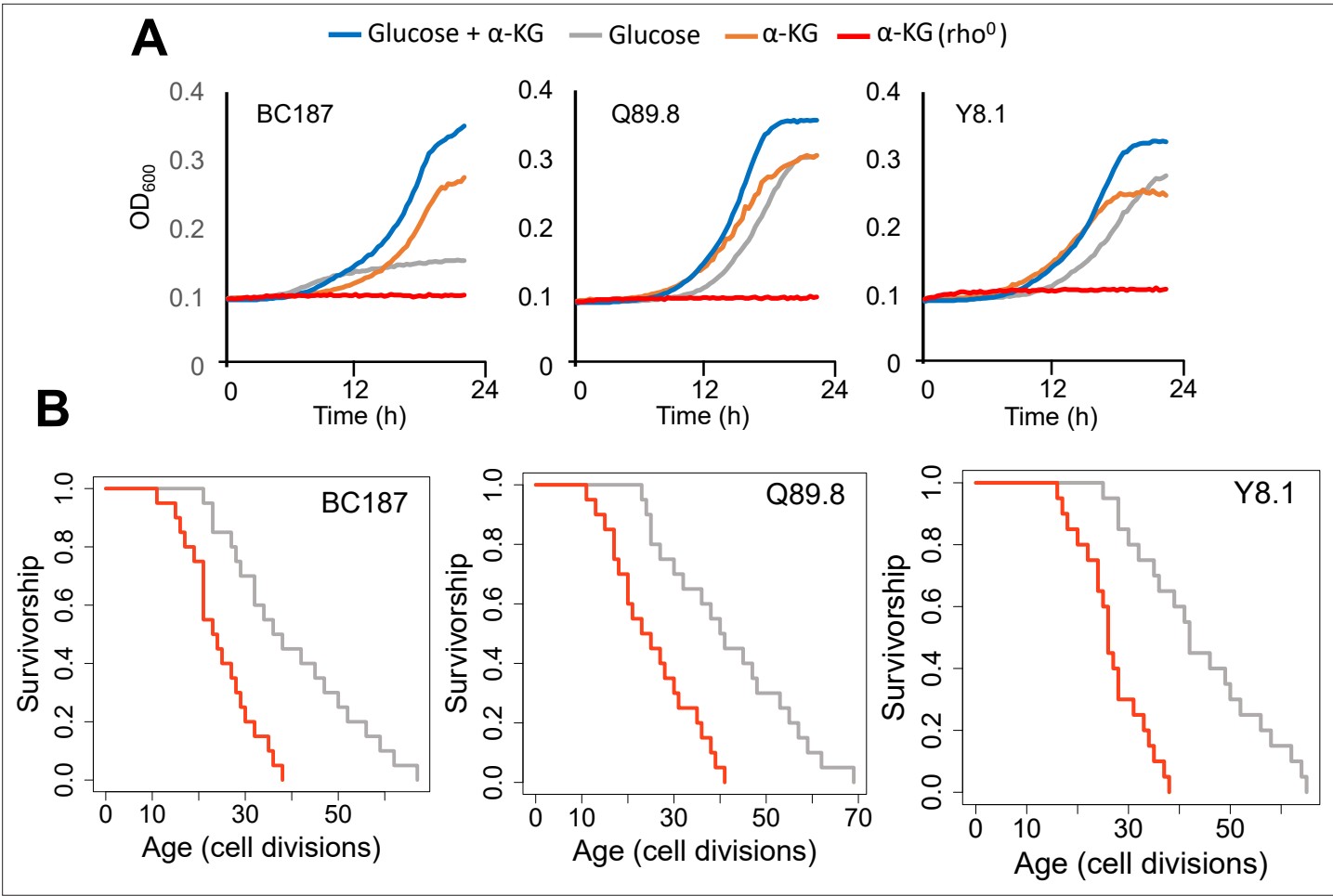

**Figure 8.** Growth properties of long-lived trains in medium supplemented with alpha-ketoglutarate (α-KG) and effect of mitochondrial DNA (mtDNA) elimination on α-KG utilization and replicative lifespan (RLS). (**A**) The growth of three long-lived strain was further supported with α-KG supplementation (10 g/l). However, strains lost the ability of α-KG utilization upon mtDNA elimination (red). Growth data of OD600 measurement can be found in *Supplementary file 1*. (**B**) Elimination of mtDNA significant reduced RLS in all three long-lived strains. Lifespan data and significance of lifespan changes can be found in *Supplementary file 1*.

The online version of this article includes the following figure supplement(s) for figure 8:

**Figure supplement 1.** Effect of alpha-ketoglutarate (α-KG) on growth.

## Discussion

The budding yeast has contributed significantly to our understanding of genetics and cell biology and has become an important model of aging, ever since Mortimer discovered the yeast RLS pheno-type (*Mortimer and Johnston, 1959*). With the power of genetics and experimental tools, yeast has provided various clues for understanding the aging process in eukaryotes and yielded hypotheses that have been further tested in other organisms, including mammals (*Kennedy, 2008*; *Fontana et al., 2010*). From this perspective, the natural isolates we analyze in the current study offer an excellent new model for yeast aging studies (*Kaya et al., 2015*; *Stumpferl et al., 2012*; *Kwan et al., 2013*; *Janssens and Veenhoff, 2016*; *Jung et al., 2018*; *Barré et al., 2020*), allowing us to leverage the enormous genetic variation found among natural isolates to study cellular processes that affect lifespan variation in nature, in a way not possible with the standard approach of deletion mutants in lab strain. In fact, these studies revealed several previously known, as well as novel, cellular processes and genetic factors that together determine RLS of yeast. For example, quantitative trait locus (QTL) analysis revealed a possible role of rDNA origin activation, nutrition sensing pathways, and serine biosynthesis in modulation of replicative and CLS in wild yeast isolates. In addition, both initial and age-associated increase in cell size found to be negatively correlated with RLS. In general, these

studies pointed out diet-dependent metabolic regulations in lifespan regulation (*Kaya et al., 2015*; *Stumpferl et al., 2012*; *Kwan et al., 2013*; *Janssens and Veenhoff, 2016*; *Jung et al., 2018*; *Barré et al., 2020*).

In this study, we further advanced these findings by utilizing -omics approaches across highly diverse aging phenotypes. Our comparison of gene expression changes and longevity signatures across laboratory-adapted long-lived mutants and long-lived natural isolates identified many genes and pathways associated with longevity. However, further studies are needed to determine their individual and collective roles in lifespan variation. At the pathway level, the transcriptomic and the metabolomic data suggest that respiratory metabolism is important for longevity, as long-lived strains are characterized with increased TCA cycle and oxidative phosphorylation activities. This is also consistent with prior data that genetic induction of respiration in the PSY316 laboratory-adapted strain is sufficient to increase RLS (*Lin et al., 2002*). In addition, we found that short-lived strains when grown with glucose as the primary carbon source (YPD) tend to achieve the largest lifespan gains when grown on glycerol (YPG) that induces a metabolic shift away from fermentation and toward respiration. In contrast, strains that are long lived on YPD generally did not show a further RLS increase. Taken together, these findings suggest possible adaptive mechanisms under glucose conditions that suggest a metabolic shift from fermentation to respiration to increase mitochondrial metabolism in long-lived isolates. Accordingly, further increase in respiration by shifting the carbon source from glucose to glycerol was not beneficial in those long-lived strains.

In addition, our combined analyses of transcriptome and metabolome data pinpointed a regulation of interconnected amino acid biosynthetic pathways, which are downregulated in long-lived strains. Amino acids are the building blocks of proteins, and it is known that individual supplementation or restriction of several different amino acids can exert both pro- and antilongevity effects mainly through a well-studied target of rapamycin (TOR) pathway (*Canfield, 2019*; *Mirzaei et al., 2014*). For example, restriction of BCAAs was shown to increase both healthspan and longevity in mice (*Wolfson et al., 2016*; *Richardson et al., 2021*). Jiang et al. first reported that reducing the amino acid content of the media can increase RLS in a short-lived laboratory yeast strain (*Jiang et al., 2000*). Similarly, Asp restriction (*Powers et al., 2006*) or treatment with the glutamine synthetase inhibitor methionine sulfoximine (*Kaeberlein et al., 2005b*) can extend RLS by inhibiting TOR. Our data are consistent with the idea that decreased amino acid biosynthesis plays a role in a longer lifespan of wild isolates and that this is associated with decreased glycolytic activity and increased mitochondrial function.

Among the amino acids that are found to be associated with lifespan, Trp appears to be particularly relevant for lifespan regulation. Recently, a decrease in KYN metabolic pathway activity through RNAi knockdown of *TDO-2* (*BNA2* ortholog) expression (knockdown) was found to robustly extend lifespan (*van der Goot et al., 2012*), while complete KO of TDO-2 expression diminished the positive lifespan effect in *C. elegans* (*Michels et al., 2016*). Increased KYN pathway activity and alteration in KYN pathway metabolites with age have also been observed in humans, suggesting a possible conserved role for this pathway in lifespan regulation (*de Bie et al., 2016*; *Chatterjee et al., 2018*). In addition, a study across 26 mammalian species found that species characterized by increased KYN pathway activity were shorter lived (*Powers et al., 2006*). In yeast, it was shown that deletion of *BNA2*, which encodes the protein that controls the first rate-limiting step in Trp catabolism, decreased RLS, while increased *BNA2* dosage (overexpression) increased RLS in diploid laboratory WT cells (*Beas et al., 2020*). We observed that transcript abundance of *BNA2* negatively correlated with lifespan across wild isolates. Consistent with this observation, we found that increased *BNA2* dosage caused a significant lifespan reduction in long-lived strains as well as short-lived strains. These data support the model that decreased KYN pathway activity is associated with increased lifespan across wild isolates. Due to decreased shikimate pathway activity, increased *BNA2* dosage possibly caused increased activation of the KYN pathway by increasing Trp degradation, which in turn resulted in decreased intracellular Trp pool for protein translation in long-lived strains. On the other hand, short-lived strains are already characterized with increased *BNA2* abundance and further increase in *BNA2* dosage might increase KYN pathway metabolic intermediates (e.g., kynurenine, QA) and result in further lifespan reduction. Perhaps, the direct way to test the role of Trp in lifespan regulation should be to analyze the effect of decreased expression of *BNA2* in short-lived strains, which will directly increase the abundance of Trp, in a similar fashion to that in long-lived strains. Although our data differ from the recently published report in which *BNA2* overexpression increased lifespan (*Beas et al., 2020*), it is possible that Trp

metabolic regulation and KYN pathway activity might follow different metabolic and genetic trajectories across wild isolates in comparison to the laboratory-adapted strain used in that study, which had been cultured on a medium with high glucose and abundant Trp over many generations.

The KYN pathway activity could also be related to $NAD^+$ homeostasis since Trp degradation is the major route for $NAD^+$ synthesis (*Choudhary et al., 2014*). Accordingly, we hypothesized that the observed unchanged $NAD^+$ abundance across yeast isolates might be explained by the increased activity of the downstream $NAD^+$ salvage pathway. In fact, we found that the expression of the nicotinamidase gene, *PNC1*, in the salvage pathway positively correlates with lifespan (*Figure 3*). Nicotinamide (NAM) is a by-product generated during Sir2p-mediated deacetylation and can be taken up from the medium. The stress-induced nicotinamidase Pnc1p in yeast is responsible for the clearance of NAM by converting it to nicotinic acid, which is a precursor for $NAD^+$ biosynthesis via the salvage pathway (*Anderson et al., 2003*; *Imai and Guarente, 2016*). Increased expression of *PNC1* alone has been shown to modulate intracellular $NAD^+$ homeostasis and to increase RLS (*Beas et al., 2020*; *Choudhary et al., 2014*). In addition to the hypothesis that the increased salvage pathway activity might compensate for $NAD^+$ biosynthesis in long-lived strains with decreased KYN activity, our finding of increased lactate abundance in these strains could be interpreted as an alternative route for $NAD^+$ regeneration. During LA fermentation, two molecules of pyruvate are converted to two molecules of LA. This reaction also supports oxidation of NADH to $NAD^+$. Previously, it has been shown in both yeast and mammalian cells that when $NAD^+$ demand is higher relative to ATP turnover, cells engage in anaerobic glycolysis, despite available oxygen (*Luengo et al., 2021*). Our data also suggest that a similar mechanism might have evolved to regulate $NAD^+$ homeostasis in cells with decreased KYN pathway activity. Overexpression of *BNA2* might also interfere with these adaptive changes in long-lived strains, which in turn decreases lifespan. Our molecular identification of adaptive metabolic changes may prove useful in uncovering additional mechanisms regulating cellular $NAD^+$ metabolism and their association with the aging process in future studies.

A potential connection between altered amino acid biosynthesis and the TCA cycle that may be particularly relevant for lifespan determination is Glu metabolism. Other than being a precursor in many amino acid biosynthetic pathways, Glu is an important carbon and nitrogen carrier, and can be catabolized to α-KG, an intermediate of the TCA cycle through a deamination reaction catalyzed by GDH2 as well as by other transaminases such as *BAT1*, *BAT2*, and *ARO8* during amino acid biosynthesis. The movement of α-KG through the TCA cycle represent the major catabolic step for the production of nucleotides, lipids, and amino acids (*Csibi et al., 2013*). Here, we also showed that wild yeast isolates can use α-KG as an alternative carbon source for growth. We hypothesize that mitochondria-specific Glu to α-KG conversion by GDH2 might be an important determinant of lifespan regulation. In fact, increasing utilization of the Glu pool toward Gln resulted in a significant decrease in lifespan in long-lived strains. Based on these data, both compartment-specific Glu to α-KG conversion by GDH activity and utilization of α-KG for energetic and/or anabolic purposes might result in longer lifespan across wild isolates. Hence, our data suggest a possible mechanism that niche-specific nutrient depletion promotes halting the biosynthetic machinery (e.g., amino acid biosynthesis and glycolysis) and alleviates catabolic processes of alternative carbon sources to provide energy maintenance in long-lived strains by increasing respiration. Recently, α-KG emerged as a master regulator metabolite (*Huergo and Dixon, 2015*). There have been many enzymes found to be regulated by α-KG, characterized as an epigenetic regulator, and identified as a regulator of lifespan in *C. elegans* (*Chin et al., 2014*) and mouse (*Asadi Shahmirzadi et al., 2020*). In *C. elegans*, α-KG was found to decrease ATP levels by blocking mitochondrial complex V activity, thereby reducing oxygen consumption. This effect was found to be mTOR dependent (*Chin et al., 2014*). Similarly, α-KG was found to extend lifespan in fruit fly by inhibiting mTOR and activating AMPK signaling (*Su et al., 2019*). In mid-aged mice, α-KG supplementation decreased systemic inflammatory cytokines leading to health and lifespan benefits (*Asadi Shahmirzadi et al., 2020*). More recently, an analysis of 178 genetically characterized inbred fly strains revealed α-KG-dependent lifespan regulation under dietary restricted conditions (*Jin et al., 2020*). However, in yeast, the effect of α-KG supplementation on lifespan regulation has never been tested. It was shown that yeast can actively transport α-KG from medium to cytosol and into mitochondria (*Casal et al., 2008*; *Zhang et al., 2020*). In addition, in contrast with the findings in *C. elegans*, α-KG supplementation was found to increase oxygen consumption in yeast (*Casal et al., 2008*). Also, α-KG supplementation was shown to increase oxidative stress resistance in

yeast (*Bayliak et al., 2018*). Similarly, a dietary role of Glu in aging has been tested in different model organisms, including yeast. Initially, Glu restriction was found to increase yeast chronological lifespan (CLS; *Wu et al., 2013*); however, later on, it was shown that Glu supplementation also has a positive effect on CLS (*Powers et al., 2006*). Similarly, in *C. elegans*, medium supplemented with a lower dose (1–5 mM) of Glu was found to extend lifespan (*Canfield, 2019*). However, the role and mechanisms of Glu metabolism in lifespan regulation are not well understood. Both α-KG and Glu are involved in epigenetic and redox regulations that all have been implicated in lifespan regulation (*Asadi Shahmir-zadi et al., 2020*; *Gregory et al., 2019*) and might also provide a mechanism for lifespan extension in long-lived strains. Furthermore, catalysis of Glu to α-KG also yields NH4, which has been shown to be involved in regulation of mTOR1 and mTOR2 signaling (*Tate and Cooper, 2003*; *Stracka et al., 2014*) and lifespan. All these findings from different organisms suggest complicated mechanisms of beneficial effects of α-KG, which needs further investigation.

Overall, our research takes advantage of natural variation in yeast lifespan that has arisen in response to mutation, selection, and genetic drift, and uses this variation to identify the potential causal roles that gene expression and metabolism play in shaping lifespan within the same species. Our data revealed a novel mechanism wherein different life-history trajectories contribute to mitochondrial metabolism. Hence, the tricarboxylic acid cycle (TCA) cycle represents a central metabolic hub to provide metabolites to meet the demands of proliferation and other cellular processes. With respect to this, modification of TCA metabolic fluxes and metabolite levels in response to environmental pressures might therefore account for cellular adaptation and plasticity in the changing environment which might also affect lifespan of these wild isolates. We further provide molecular insights into the unique metabolic adaptation involving linked pathways, involving in Glu and α-KG metabolisms in regulation of mitochondrial function and their possible association with lifespan variation. Further understanding of how gene–environment interactions modulates genes and pathways associated with longevity may open new therapeutic applications to slow aging and delay the onset of age-related diseases through diet, lifestyle, or pharmacological interventions. In future studies, it might yield important information to investigate the role of α-KG metabolism in amino acid and caloric restricted lifespan regulation.

## Materials and methods
### Yeast strains and growth conditions

Many of the diploid wild isolates of *S. cerevisiae* and *S. paradoxus* (68 isolates) were obtained from the Sanger Institute (*Liti et al., 2009*) and the remaining 8 isolates of *S. cerevisiae* were gifted by Justin Fay from Washington University (*Hyma and Fay, 2013*). Detailed information about strains used in this study is in *Supplementary file 1*. The diploid laboratory WT strain BY4743 was purchased from the American Type Culture Collection. For testing the growth effect, strains were cultured overnight in a 96-well plate incubator at 30 °C in YPD medium. Next day, 1 µl from overnight culture was transferred to the YP, YP + α-KG (10 g/l), or YPD (0.02 % glucose) + α-KG and growth was monitored in 96-well plate using Epoch2 (BioTek, Winooski, VT, USA) kinetic growth analyzer by analyzing optical density of $OD_{600}$. For expression of genes of interest, we used modified p426GPD high copy plasmid by inserting a hygromycin (HYG) cassette along with its promoter and terminator at the *XbaI* restriction site. HYG cassette was amplified from pGAD32 plasmid with PCR. Using modified p426GPD, we inserted *GLN1* and *BNA2* gene cassettes individually at the *BamHI/XhoI* restriction sites for overexpression. Yeast transformation was performed using standard lithium acetate method. Growth rates were determined using a BioScreen-C instrument (Bioscreen C MBR, Piscataway, NJ, USA) by the analysis of optical density in the $OD_{600}$ range, and doubling times were calculated with an R script by analyzing fitting spline function from growth curve slopes (*Kahm et al., 2010*). The maximum slope of the spline fit was used as an estimate for the growth rate and doubling time for each evolved line, in combination with the YODA software package (*Olsen et al., 2010*). Finally, mtDNA was eliminated by culturing cells in YPD medium, supplemented with 10 µg/ml and ethidium bromide (EtBr). Briefly, logarithmically growing cells ($OD_{600}$ = 0.5) were incubated at room temperature with agitation for approximately 24 hr. Following a second and third treatment with the same concentration of EtBr for 24 hr, the cells were diluted (1:100) in water and plated on YPD to obtain single colonies. After then, several individual colonies were selected for testing their growth ability on YPG plates. Colonies, which were unable to grow on YPG were selected as rho[0].

## RLS assay

RLS was determined using a modification of our previously published protocol (*Steffen et al., 2009*). Yeast cell cultures for each strain were freshly started from frozen stocks on YPD plates and grown for 2 days at 30 °C prior to dissections. Several colonies were streaked onto new YPD with 2 % glucose, YPD with 0.05 % glucose, or YPG plates with 3 % glycerol using pipette tips. After overnight growth, ~100 dividing cells were lined up. After the first division, newborn daughter cells were chosen for RLS assays using a dissection microscope. For each natural isolate, at least two independent assays were performed using at least sets of 20 cells for each assay. Each assay included 20–80 mother cells of BY4743 strain as well, which was used in every experiment as a technical control. For RLS analysis of wild isolates harboring expression plasmids, individual colonies were picked up from selection medium (HYG) and YPD medium supplemented with 200 µg/ml HYG were used for RLS determination of these cells. Survival analysis and Gompertz modeling were performed using the survival and flexsurv packages in R, respectively.

## Measurement of basal OCR and western blot analysis

To investigate metabolic respiration differences across wild isolates OCR (pmol/min) was measured using a Seahorse XFe96 analyzer (Agilent, Santa Clara, CA, USA). Cells grown overnight in YPD were diluted to OD600 = 0.01 in the morning, and cells were grown to reach the OD600 = 0.25–0.5. Then, cell culture was diluted to OD600 = 0.02 in YPD and placed in a XFe96 cell culture plate coated with 15 µl 0.01 % poly-L-lysine and attached to the plate according to the previously published protocol (*Lev et al., 2020*). Basal OCR was measured for five cycles at 30 °C. To examine the expression of mitochondrial proteins, western blotting was carried out with antibodies against mitochondrial outer membrane protein Por1 (Abcam, Cambridge, MA, USA, cat:ab110326). For each strain, 10 ml logarithmically growing cells were collected and proteins were isolated according to previously published protocol (*Kaya et al., 2015*). The membranes were stripped and developed with antibodies against phosphoglycerate kinase (Pgk1; Life Technologies, Grand Island, NY, USA, cat: 459250) as an internal loading control.

## RNA-sequencing and data analysis

Three independent cultures for each strain were collected at the $OD_{600}$ = 0.4 on YPD medium to isolate RNA from each culture using Quick-RNA 96 Kit from Zymo Research (Cat. number: R1053). To prepare RNA-seq libraries, Illumina TruSeq RNA library preparation kits were used according to the user manual, and RNA-seq libraries were loaded on Illumina HiSeq 4000 platform to produce 150 bp paired-end sequences. After quality control and adapter removal, the STAR software package (*Dobin and Gingeras, 2015*) was used to map the reads against a pseudo reference genome of each strain, in which we replaced identified nucleotide changes in the S288c reference genome. Read alignment rate for transcriptome data against pseudo genome varied between 92% and 97% across *S. cerevisiae* strains and 93% and 99% across *S. paradoxus* strains (*Figure 2—figure supplement 1*). Read counts per gene were calculated using featureCounts (*Liao et al., 2014*). To filter out genes with low numbers of reads, we used filterByExpr function from the edgeR package and resulted in an expression set of 5376 genes across replicates of wild isolates.

## Metabolite profiling and data analysis

A portion of the cell pellet collected for RNA-seq analyses was also used for targeted metabolite profiling using LC–MS. 1 ml of MeOH:$H_2O$ mixture (8:2, vol/vol) was added to the samples, swirled at 550 rpm on a mixer for 5 min and then transferred to an Eppendorf tube, they were sonicated in an ice bath for 10 min, centrifuged at 4 °C at 14,000 rpm for 15 min, and 600 µl of supernatant was collected into a new tube and dried in a vacuum centrifuge at 30 °C for 2.5 hr. Samples were reconstituted in 1 ml and injected into a chromatography system consisting of a dual injection valve setup allowing injections onto two different LC columns with each column dedicated to an ESI polarity. 5 µl were injected on the positive mode column and 10 µl on the negative side column. The columns were a matched pair from the same production lot number and were both a Waters BEH amide column (2.1 × 150 mm). Auto sampler was maintained at 4 °C and column oven was set to 40 °C. Solvent A 95 % $H_2O$, 3 % acetonitrile, 2 % methanol, 0.2 % acetic acid with 10 mM ammonium acetate and 5 µM medronic acid, and Solvent B (5 % H2O, 93 % acetonitrile, 2 % methanol, 0.2 % acetic acid with

10 mM ammonium acetate 5 µM medronic acid) were used for sample loading. After completion of the 18 min gradient, injection on the opposite column was initiated and the inactive column was allowed to equilibrate at starting gradient conditions. A set of QC injections for both instrument and sample QC were run at the beginning and end of the sample run. Data were integrated by Multiquant 3.0.2 software. Peaks were selected based on peak shape, a signal-to-noise of 10 or better and retention times consistent with previously run standards and sample sets. Analysis of the dataset was performed using R (version 3.6.0). All the metabolites with ≥40 % missingness were excluded, and a total of 166 metabolites were included in the imputation step. We imputed the remaining missing values using the *K*-nearest neighbors imputation method implemented in the R impute package. The log2-transformed abundance was Cyclic LOESS normalized prior to imputation.

## Principal component analysis

PCA was performed on preprocessed data (e.g., normalized and imputed log2 abundance of the metabolomic data, and the log2-counts per million [CPM] values of the filtered and TMM normalized RNA-seq data) using the R prcomp function. To identify the underlying pathways, the factors in each of the first three PCs were ranked by their contributions, and pathway enrichment analysis was performed on the top 500 transcripts using Network Analyst (*Zhou et al., 2019*) and on the top 40 metabolites using MetaboAnalyst (*Pang et al., 2021*) platforms.

## Phylogenetic regression by generalized least squares

R packages 'nmle' and 'phylolm' were used to perform phylogenetic regression by generalized least squares method to identify RLS association of transcripts and metabolites (*Kaya et al., 2015*). We tested four models of trait evolution: (1) complete absence of phylogenetic relationship ('Null'); (2) Brownian Motion model ('BM'); (3) BM transformed by Pagel's lambda ('Lambda'); and (4) Ornstein–Uhlenbeck model ('OU'). The parameters for Lambda and OU models were estimated simultaneously with the coefficients using maximum likelihood. The best-fit model was selected based on maximum likelihood. Strength of correlation was based on the p value of regression slope. To confirm robustness of results, regression was performed by leaving out each strain, one at a time, and computing p values using the remaining strains.

## Gene expression signature associated with RLS across deletion strains

Gene expression data on deletion mutants was obtained from GSE45115, GSE42527, and GSE42526 (*Kemmeren et al., 2014*). The corresponding RLS lifespan data for mutant strains were from *McCormick et al., 2015*. Based on the raw data from the number of replicates, we calculated median, mean, and maximum RLS, together with corresponding standard errors for each deletion strain. In total, this resulted in 1376 deletion strains, for which both RLS and gene expression data were available. logFC of individual genes corresponding to each mutant strain compared to control samples were used for subsequent analysis.

To identify genes associated with RLS across KO strains linear models in limma were used (*Ritchie et al., 2015*). We found genes associated with median, mean, and maximum RLS both in linear and logarithmic scale, and BH adjustment was performed to account for multiple hypotheses (*Benjamini and Hochberg, 1995*). Genes with adjusted p value <0.05 were considered significant. To determine statistical significance of the overlap between genes associated with different metrics of RLS, we performed Fisher's exact test separately for up- and downregulated genes, considering 6170 genes as background.

## Comparison between signatures of RLS across deletion and natural strains

To compare gene expression signatures associated with different metrics of RLS across deletion and natural *Saccharomyces* strains, we calculated Spearman correlation coefficients between corresponding gene expression slope coefficients in a pairwise manner. Clustering of the Spearman correlation matrix was performed with the complete hierarchical approach.

To increase the signal within the correlation matrix, the union of top 1000 statistically significant genes from each of the 2 signatures in a pair was used to calculate Spearman correlation coefficient. To get an optimal gene number for removal of noise, we looked at how the total number of significantly correlated pairs of signatures depended on the number of genes used to calculate the

correlation coefficient. As a threshold for significance, we considered BH adjusted p value <0.05 and Spearman correlation coefficient >0.1.

To determine statistical significance of the overlap between transcripts associated with different metrics of RLS across deletion and natural strains, we performed Fisher's exact test, considering 4712 genes as background. To identify genes whose deletions are associated with longer or shorter lifespan in *S. cerevisiae* strains, we compared the distribution of RLS across samples corresponding to certain deletion strains with the distribution of median RLS across all measured deletion strains. For that we used Mann–Whitney test. Genes with BH adjusted p value <0.05 were considered significant. Overlap of these genes with lifespan-associated genes across natural strains was assessed with the Fisher's exact test with BH adjusted p value threshold of <0.05.

## Functional enrichment analysis

For the identification of functions enriched by genes associated with RLS across deletion and natural strains, we performed GSEA (*Subramanian et al., 2005*) on a ranked list of genes based on $\log_{10}$(p value) corrected by the sign of regulation, calculated as:

$$- (pv) \times sgn (b)$$

where *pv* and *b* are p value and slope of expression of a certain gene, respectively, and *sgn* is signum function (is equal to 1, −1, and 0 if value is positive, negative, and equal to 0, respectively). REACTOME, KEGG, and gene ontology (GO) biological process from Molecular Signature Database (MSigDB) were used as gene sets for GSEA (*Liberzon et al., 2011*). We utilized the fgsea package in R for GSEA analysis. Adjusted p value cutoff of 0.1 was used to select statistically significant functions. We visualized several manually chosen statistically significant functions with a heatmap colored based on normalized enrichment score. Clustering of functions has been performed with hierarchical complete approach and Euclidean distance. Combined integrative analysis of transcriptomics and metabolomics data for pathway analysis was performed by using joint-pathway analysis option in MetabolAnalyst 5.0 (*Pang et al., 2021*).

# Additional information

## Competing interests

Matt Kaeberlein: Reviewing editor, *eLife*. The other authors declare that no competing interests exist.

## Funding

| Funder | Grant reference number | Author |
|---|---|---|
| National Institutes of Health | 1K01AG060040 | Alaattin Kaya |
| National Institutes of Health | AG067782 | Vadim N Gladyshev |
| National Institutes of Health | AG064223 | Vadim N Gladyshev |
| National Institutes of Health | AG049494 | Daniel EL Promislow |
| National Institutes of Health | T32 AG052354 | Mitchell Lee |
| University of Washington | P30AG013280 | Alaattin Kaya |

The funders had no role in study design, data collection and interpretation, or the decision to submit the work for publication.

## Author contributions

Alaattin Kaya, Conceptualization, Data curation, Formal analysis, Funding acquisition, Investigation, Methodology, Project administration, Resources, Supervision, Validation, Visualization, Writing – original draft, Writing – review and editing; Cheryl Zi Jin Phua, Formal analysis, Methodology,

Validation, Visualization; Mitchell Lee, Formal analysis, Methodology, Validation, Visualization, Writing – original draft, Writing – review and editing; Lu Wang, Formal analysis, Methodology, Software, Validation, Visualization, Writing – original draft, Writing – review and editing; Alexander Tyshkovskiy, Formal analysis, Methodology, Software, Validation, Visualization, Writing – original draft; Siming Ma, Data curation, Formal analysis, Methodology, Software, Validation, Visualization, Writing – original draft; Benjamin Barre, Methodology, Writing – review and editing; Weiqiang Liu, Data curation, Formal analysis, Methodology, Software, Validation, Writing – original draft; Benjamin R Harrison, Conceptualization, Methodology, Writing – original draft, Writing – review and editing; Xiaqing Zhao, Formal analysis, Writing – review and editing; Xuming Zhou, Formal analysis, Writing – original draft; Brian M Wasko, Theo K Bammler, Formal analysis, Validation; Daniel EL Promislow, Formal analysis, Validation, Writing – original draft, Writing – review and editing; Matt Kaeberlein, Formal analysis, Methodology, Supervision, Validation, Visualization, Writing – original draft, Writing – review and editing; Vadim N Gladyshev, Funding acquisition, Investigation, Project administration, Supervision, Validation, Writing – original draft, Writing – review and editing

**Author ORCIDs**
Alaattin Kaya  http://orcid.org/0000-0002-6132-5197
Matt Kaeberlein  http://orcid.org/0000-0002-1311-3421
Vadim N Gladyshev  http://orcid.org/0000-0002-0372-7016

**Decision letter and Author response**
Decision letter https://doi.org/10.7554/eLife.64860.sa1
Author response https://doi.org/10.7554/eLife.64860.sa2

## Additional files

**Supplementary files**
• Supplementary file 1. Strain list, used in this study and their replicative lifespan along with doubling time are listed. This file also includes replicative lifespan of the laboratory KO strains, analyzed in this study.

• Supplementary file 2. File includes raw and normalized counts from RNA-seq. Normalized metabolite count is also included. Results from phylogenetic generalized least square (PGLS) analysis from both datasets along with statistical values are listed for natural isolates. Similarly, genes associated with RLS (mean, median, and maximum) of KO strains along with statistical values are also listed. This file also includes result GSE enrichment terms with statistical values for natural isolates and KO strains.

• Supplementary file 3. PCA loadings for normalized RNA-seq and metabolomics datasets.

• Supplementary file 4. List of yeast longevity genes from GenAge database. Identified set of CR-related genes from our analysis is also listed.

• Supplementary file 5. Values for oxygen consumption rate (OCR) measurement across the strains.

• Transparent reporting form

**Data availability**
All data generated or analyzed during this study are included in the manuscript and supporting files. RNA-seq data were deposited with the NCBI Gene Expression Omnibus (GEO) with accession number GSE188294.

The following dataset was generated:

| Author(s) | Year | Dataset title | Dataset URL | Database and Identifier |
|---|---|---|---|---|
| Alaattin K, Siming M, Xuming Z | 2021 | Evolution of Natural Lifespan Variation and Molecular Strategies of Extended Lifespan | https://www.ncbi.nlm.nih.gov/geo/query/acc.cgi?acc=GSE188294 | NCBI Gene Expression Omnibus, GSE188294 |

The following previously published datasets were used:

| Author(s) | Year | Dataset title | Dataset URL | Database and Identifier |
|---|---|---|---|---|
| Kemmeren P, Sameith K, van de Pasch L, Holstege F | 2014 | Expression profiling of 376 wildtypes to assess day-to-day variance | https://www.ncbi.nlm.nih.gov/geo/query/acc.cgi?acc=GSE45115 | NCBI Gene Expression Omnibus, GSE45115 |
| Kemmeren P, Sameith K, van de Pasch L, Holstege F | 2014 | Compendium of deletion mutant gene expression profiles, 700 responsive mutants [hs1991] | https://www.ncbi.nlm.nih.gov/geo/query/acc.cgi?acc=GSE42527 | NCBI Gene Expression Omnibus, GSE42527 |
| Kemmeren P, Sameith K, van de Pasch L, Holstege F | 2014 | Compendium of deletion mutant gene expression profiles, 784 non-responsive mutants [hs1990] | https://www.ncbi.nlm.nih.gov/geo/query/acc.cgi?acc=GSE42526 | NCBI Gene Expression Omnibus, GSE42526 |

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
