## [Editor Report]

This manuscript characterizes differences in replicative lifespan across a collection of natural yeast isolates. Additionally, it provides transcriptional and metabolomic resources that help shed important new light on the mechanistic bases of functional differences among the wild yeast isolates. The revised version further tests the impact on lifespan in short- and long-lived strains of target genes identified. This work has important implications for the fields biology of aging and potentially evolutionary biology.

---

## [Decision Letter]

**Decision letter after peer review:**

Thank you for submitting your article "Evolution of Natural Lifespan Variation and Molecular Strategies of Extended Lifespan" for consideration by *eLife*. Your article has been reviewed by 2 peer reviewers, and the evaluation has been overseen by a Reviewing Editor and Jessica Tyler as the Senior Editor. The reviewers have opted to remain anonymous.

Essential revisions:

Overall, as noted by both reviewers, experimental validation of the causal basis of lifespan differences between strains or species would improve the rigor of the work.

1) For example, the short-chain fatty acids and/or amino acids identified to correlate or anti-correlate with life-span are connected also to several other of the hallmarks of aging/aging pathways identified in baker's yeast (Lopez-Otin, Cell, 2013; Janssens G, Microb Cell, 2016) and it would benefit the reader if the authors tried to give a more balanced overview of interacting mechanisms.

2) The omics data lead up to the observations that tryptophan positively and kynurenine pathway activity negatively correlates with life-span. These data would be significantly strengthened if the authors could support their importance in life-span regulation by genetic means. Other yeast aging studies using natural isolates have used reciprocal hemizygosity to delete/switch QTLs identified by genome-wide association studies to differ between strains (eg Stumpferl S, Genome Res, 2012; Kwan EX, PLOS Genet, 2013; Barré B, Genome Res, 2020). Although the genetic resolution in this small size collection of rather divergent natural isolates may not narrow down the changes needed into highly targeted genetic alterations, the genes and/or activities of pathways suggested to contribute to aging could be ablated/overexpressed to test the ideas proposed. For example, it would be interesting to know what the impact of BNA2 ablation/overexpression or alterations of other genes in the kynurenine pathway is on the replicative life-span of selected strains, or of PNC1 or genes in lysine biosynthesis are on their longevity. Without proper testing of hypotheses the study remains speculative and although interesting is better suited for a more specialized journal.

*Reviewer #1:*

This manuscript seeks to characterize and dissect differences in replicative lifespan between wild yeast. The experiments are a tour de force (laborious single-cell lifespan assays done by hand plus transcriptome and metabolome data) and appear to be carefully done. This group previously published lifespans of 22 wild *S. cerevisiae* and their correlation with published transcriptome, proteomic, and metabolomic data (ref. 18, PMID 27030810). Here they add ~20 more plus strains of another species, S. paradoxus.

The two most striking and novel results of this manuscript are the following. First, Figure 1 shows that S. paradoxus are longer-lived than *S. cerevisiae*. Species-level divergence are much more likely to be driven by natural selection than those across a population, and thus this result hints at a story about the evolution of yeast lifespan which could be of very high impact indeed. Second, Figure 4 shows that transcripts correlating with lifespan in panels of lab-generated mutants are completely disjoint from those discovered among wild strains here. This attests to the utility of the current data as a contrast from the literature to date in laboratory yeast.

Beside the overall distinction in lifespan between species, Figure 1 also shows that forcing respiratory state (via growth on glycerol) extends lifespan across genotypes whereas caloric restriction only benefits a subset of genotypes. The authors highlight the agreement with trends they have seen previously in lab-generated mutants rather than wild isolates.

Figure 2 reports global analyses of strain transcriptomes and metabolomes, which are robustly different between the species.

Figures 3 and 5 show the transcripts and metabolites, respectively, that correlate most strikingly with lifespan across wild strains and species (mirroring previous analyses in ref. 18, but much stronger relationships are seen here).

Figure 6 gives gene modules that distinguish the transcriptomes of the strains that benefit from caloric restriction.

Experimental validation of the causal basis of lifespan differences between strains or species would improve the rigor of the work.

Figure 2 is largely a description of expression differences between species, of which most may be irrelevant for lifespan, and could be eliminated.

Reducing speculation in the text would help readability (e.g. interpretation of metabolites correlated with lifespan which have not been subject to experimental validation and comprise pages 13-15).

*Reviewer #2:*

The authors set out to elucidate the mechanisms underlying the natural variation in replicative aging in a small set of naturally isolated closely related yeast strains. As such the study addresses the question of mechanisms explaning natural variation in longevity, which indeed may be interesting for a more general audience. The low number of yeast strains assessed likely prevented the use of genome-wide association studies (GWAS), a frequently used method to address such questions. GWAS would, with the proper number of strains and/or hybrid crosses in between them, likely have been able to point out candidate genes/polymorphisms responsible for such differences. Instead the authors investigated transcriptomic and metabolomic profiles in the strains to pin-point mechanisms involved. Whereas they were indeed via measurements of transcript and metabolite levels able to put forth a low number of pathways that may be involved, the authors unfortunately stay at mechanistic indications without proper hypothesis testing. In its present state the manuscript thus does not quite manage to fulfil the aims.

The manuscript by Kaya A et al. contains a number of interesting observations and correlations pertaining to replicative life-span in a collection of closely related natural yeast isolates. The dataset contains a number of interesting observations such as that many natural isolates appear to impinge on a similar aging pathway as caloric restriction. It is generally well written, although the discussions around the data are rather one-sided.

For example, the short-chain fatty acids and/or amino acids identified to correlate or anti-correlate with life-span are connected also to several other of the hallmarks of aging/aging pathways identified in baker's yeast (Lopez-Otin, Cell, 2013; Janssens G, Microb Cell, 2016) and it would benefit the reader if the authors tried to give a more balanced overview of interacting mechanisms. Yet the authors choose to focus discussions entirely around speculations on their interconnections with NAD+/histone modifications/deacetylation and citing their own work on the topic.

Furthermore, despite its many interesting correlations, my main concern is that the manuscript lacks experimental verification of the most interesting associations made. For example, the omics data lead up to the observations that tryptophan positively and kynurenine pathway activity negatively correlates with life-span. These data would be significantly strengthened if the authors could support their importance in life-span regulation by genetic means. Other yeast aging studies using natural isolates have used reciprocal hemizygosity to delete/switch QTLs identified by genome-wide association studies to differ between strains (eg Stumpferl S, Genome Res, 2012; Kwan EX, PLOS Genet, 2013; Barré B, Genome Res, 2020). Although the genetic resolution in this small size collection of rather divergent natural isolates may not narrow down the changes needed into highly targeted genetic alterations, the genes and/or activities of pathways suggested to contribute to aging could be ablated/overexpressed to test the ideas proposed. For example, it would be interesting to know what the impact of BNA2 ablation/overexpression or alterations of other genes in the kynurenine pathway is on the replicative life-span of selected strains, or of PNC1 or genes in lysine biosynthesis are on their longevity. Without proper testing of hypotheses the study remains speculative and although interesting is better suited for a more specialized journal.

Regarding, BNA2 a recent study showed that its overexpression extends life-span (Beas AO, Nat Commun, 2020), thus the opposite of what could be expected based on this report. Furthermore, BNA2 overexpression and increased activity in the kynurenine pathway affected life-span without impacting on NAD+ levels, but through an as of yet poorly understood mechanism increasing the levels of kynurenine branch point metabolites e.g. xanthurenate (Beas AO, Nat Commun, 2020). Notably, the study furthermore identified age-related accumulation of lipid droplets in yeast, that were inhibited by BNA2 overexpression, an observation that may relate to the positive correlation between lipid metabolic process and replicative longevity in this study and its anti-correlation to kynurenine pathway metabolites. In fact the manuscript at hand supports the importance of kynurenine pathway metabolites independently of NAD+, since NAD+ levels do not correlate with longevity, yet the authors focus all their discussion of mechanism around NAD+ and histone deacetylases. The observation that NADH levels correlate with replicative longevity is brushed away with a discussion about technical problems in assessing their levels faithfully in vivo (p14, lines 367-375).

*Reviewer #3:*

The use of the natural diversity to dissect the genetic basis and forces acting on given trait variation is now a straightforward strategy. In this paper, the authors used it to have a better insight into the lifespan variation in yeast. For that purpose, they focused 76 wild yeast isolates coming from two different species, *Saccharomyces cerevisiae* (n=40) and Saccharomyces paradoxus (n=36). More precisely, they assayed replicative lifespan (RLS) of this collection of strains under three different conditions. This obviously allowed to observe a variation of the RLS. In parallel, they also characterized gene expression variation (transcriptomics) and differences in their metabolite levels (metabolomics) for this set of isolates. Using these datasets, they finally explored the relationship between gene expression, metabolite abundance, and longevity to determine the molecular patterns associated with lifespan variation.

This is an interesting work dealing with key aspects of lifespan variation. The question is clearly stated and analyses provide some results. This study tackles an important biological question and a new insight is brought. However, the used strategy, i.e. exclusively based on the correlation of datasets, present some caveats and some experimental validation need to be brought to really trust the conclusions.

Here, just a few points that I would like to highlight:

– One of my main concerns is the strategy that was used by the author in order to dissect this given trait, which is longevity. It is probably fair to try to find some correlations between different datasets (even if there are not totally related) but not sure the conclusions should be trusted without additional experimental validation. As clearly mentioned in the 'Material and Methods' section, cells were grown for 2 days at 30°C for the RLS assay. By contrast, strains were collected at OD = 0.4 for the RNAseq and metabolite profiling. Obviously, this makes a serious difference. Again, it's fair to compare the datasets but not sure, general conclusions can be drawn without experimental validations.

– Line 85

'thus, their natural lifespan variation must be encoded in their respective genomes'

As the authors explored the different traits with no-environmental changes in the lab, this is a very strange conclusion/statement.

– Line 107

'There was also a group of pathogenic *S. cerevisiae* strains isolated from immunocompromised patients'

*S. cerevisiae* is not a pathogen. These strains are called 'clinical isolates'.- Line 109 – 111

'To understand the forces that shape diversity of lifespan and the associated mechanisms, we first sequenced and assembled the genomes of these 76 isolates. Consistent with a previously published report [49], we found substantial genetic variation among these strains (~655,000SNPs).'

The genome of more than 2,000 *S. cerevisiae* isolates were now sequenced. Were these isolates not sequenced yet. The authors mentioned the consistency of their results with a study that was published 12 years ago and dealing with Sanger sequenced genome at a coverage of around 3x. How can that be consistent? There is a quantum leap in data quality and precision there… no real comparison seems possible and insightful. Much of the yeast population genomic literature is missing here!

– Surprisingly, one study that explored for the first time intra- and interspecies RLS variation in yeast is not cited?!?!? Jung et al. 2015 PLoS one.

– Line 220

'The resulting topology was largely consistent with their phylogeny with a clear separation between *S. cerevisiae* and S. paradoxus strains (Figure 2A).' What does 'largely consistent' mean? At the intra-specific level (*S. cerevisiae*), the topology is not consistent at all with the genomic data.

– The 'gene expression and metabolite…' paragraph is too wordy and poorly structured. It probably needs to be rewritten.

– The conclusion begins with a lengthy commentary on laboratory strain S288c. This part does not add anything and seems a little off topic.

---

## [Author Response]

Essential revisions:Overall, as noted by both reviewers, experimental validation of the causal basis of lifespan differences between strains or species would improve the rigor of the work.1) For example, the short-chain fatty acids and/or amino acids identified to correlate or anti-correlate with life-span are connected also to several other of the hallmarks of aging/aging pathways identified in baker's yeast (Lopez-Otin, Cell, 2013; Janssens G, Microb Cell, 2016) and it would benefit the reader if the authors tried to give a more balanced overview of interacting mechanisms.

Thank you. We further analyzed our data and added 3 new figures (Figure 6,7,8) and 5 new supplementary figures (Figure S5, S7, S8, S9, S10). Based on our new findings, we revised and extended the discussion to offer a more balanced, thorough and broader overview of interacting mechanisms. We also revised the introduction. Although additional future studies may be needed, we believe that these new analyses and revisions substantially improved the manuscript and narrowed down lifespan regulating mechanisms in wild yeast isolates.

2) The omics data lead up to the observations that tryptophan positively and kynurenine pathway activity negatively correlates with life-span. These data would be significantly strengthened if the authors could support their importance in life-span regulation by genetic means. Other yeast aging studies using natural isolates have used reciprocal hemizygosity to delete/switch QTLs identified by genome-wide association studies to differ between strains (eg Stumpferl S, Genome Res, 2012; Kwan EX, PLOS Genet, 2013; Barré B, Genome Res, 2020). Although the genetic resolution in this small size collection of rather divergent natural isolates may not narrow down the changes needed into highly targeted genetic alterations, the genes and/or activities of pathways suggested to contribute to aging could be ablated/overexpressed to test the ideas proposed. For example, it would be interesting to know what the impact of BNA2 ablation/overexpression or alterations of other genes in the kynurenine pathway is on the replicative life-span of selected strains, or of PNC1 or genes in lysine biosynthesis are on their longevity. Without proper testing of hypotheses the study remains speculative and although interesting is better suited for a more specialized journal.

Thank you for these comments. As the reviewers pointed out, our sample size is small for QTL analysis. Accordingly, we used endophenotypes to fill the gap between genotype and phenotype (lifespan). We comprehensively analyzed our findings and performed additional experiments including testing the lifespan effect of BNA2 overexpression in selected short- and long-lived strains (Figure 10). We also examined the consequences of overexpression of GLN1. Interestingly, expression of these genes in long-lived strains shortened lifespan, whereas in short lived strains, the lifespan was typically increased or not affected. This finding further highlights the disparate lifespan affecting mechanisms that may exist in wild isolates and their differences with those of lab strains. Based on new data of growth on alpha-ketoglutarate and lifespan of rho0 strains, we have been able to narrow down possible mechanisms associated with lifespan variation across wild isolates.

Reviewer #1:[…] Experimental validation of the causal basis of lifespan differences between strains or species would improve the rigor of the work.

We thank the reviewer for this suggestion. We tested the effects of overexpression of two genes that emerged from our studies, GLN1 and BNA2, in various long- and short-lived strains. This point was further addressed in the Essential Revisions Request #2; please see our response above.

Figure 2 is largely a description of expression differences between species, of which most may be irrelevant for lifespan, and could be eliminated.

Thank you. We agree that Figure 2 is not fully relevant to lifespan variation. However, with the new discussion added, we believe that this figure is necessary to show inter and intra species endophenotype differences. This figure is also used to illustrate our hypothesis that species and strain specific life history trajectories may affect endophenotypes which ultimately modulate lifespan. We substantially revised the description of this figure to make it more relevant to the study.

Reducing speculation in the text would help readability (e.g. interpretation of metabolites correlated with lifespan which have not been subject to experimental validation and comprise pages 13-15).

We have now performed additional experiments and modified our language throughout the manuscript, with focus on reducing speculation, to address this reviewer’s concern.

Reviewer #2:The authors set out to elucidate the mechanisms underlying the natural variation in replicative aging in a small set of naturally isolated closely related yeast strains. As such the study addresses the question of mechanisms explaning natural variation in longevity, which indeed may be interesting for a more general audience. The low number of yeast strains assessed likely prevented the use of genome-wide association studies (GWAS), a frequently used method to address such questions. GWAS would, with the proper number of strains and/or hybrid crosses in between them, likely have been able to point out candidate genes/polymorphisms responsible for such differences. Instead the authors investigated transcriptomic and metabolomic profiles in the strains to pin-point mechanisms involved. Whereas they were indeed via measurements of transcript and metabolite levels able to put forth a low number of pathways that may be involved, the authors unfortunately stay at mechanistic indications without proper hypothesis testing. In its present state the manuscript thus does not quite manage to fulfil the aims.

Thank you for the thorough assessment of our study. We extensively revised it and hope it has been substantially improved. The reviewer is right about the low number of strains for GWAS. We have actually carried out GWAS studies, but the sample size precluded reliable findings, so we did not include these analyses in the manuscript. There have been very few studies addressing natural variation in lifespan, as opposed to numerous studies in lab strains examining single gene and environmental manipulations. As such, our study fills this gap, and while we could not provide all the answers, it represents a major effort in determining how nature regulates species lifespan and does so more efficiently than any manipulations that have been examined in lab strains.

The manuscript by Kaya A et al. contains a number of interesting observations and correlations pertaining to replicative life-span in a collection of closely related natural yeast isolates. The dataset contains a number of interesting observations such as that many natural isolates appear to impinge on a similar aging pathway as caloric restriction. It is generally well written, although the discussions around the data are rather one-sided.For example, the short-chain fatty acids and/or amino acids identified to correlate or anti-correlate with life-span are connected also to several other of the hallmarks of aging/aging pathways identified in baker's yeast (Lopez-Otin, Cell, 2013; Janssens G, Microb Cell, 2016) and it would benefit the reader if the authors tried to give a more balanced overview of interacting mechanisms. Yet the authors choose to focus discussions entirely around speculations on their interconnections with NAD+/histone modifications/deacetylation and citing their own work on the topic.

Thank you. We have put major effort to offer a more balanced and thorough discussion of our findings, while not overinterpreting the findings and reducing speculation. We still preserve some of the NAD+ discussion, but include additional examples. This comment is further addressed in Essential Revisions request #1.

Furthermore, despite its many interesting correlations, my main concern is that the manuscript lacks experimental verification of the most interesting associations made. For example, the omics data lead up to the observations that tryptophan positively and kynurenine pathway activity negatively correlates with life-span. These data would be significantly strengthened if the authors could support their importance in life-span regulation by genetic means. Other yeast aging studies using natural isolates have used reciprocal hemizygosity to delete/switch QTLs identified by genome-wide association studies to differ between strains (eg Stumpferl S, Genome Res, 2012; Kwan EX, PLOS Genet, 2013; Barré B, Genome Res, 2020). Although the genetic resolution in this small size collection of rather divergent natural isolates may not narrow down the changes needed into highly targeted genetic alterations, the genes and/or activities of pathways suggested to contribute to aging could be ablated/overexpressed to test the ideas proposed. For example, it would be interesting to know what the impact of BNA2 ablation/overexpression or alterations of other genes in the kynurenine pathway is on the replicative life-span of selected strains, or of PNC1 or genes in lysine biosynthesis are on their longevity. Without proper testing of hypotheses the study remains speculative and although interesting is better suited for a more specialized journal.

Thank you. We used the reviewer’s suggestion and tested genes identified by our analyses. Specifically, we tested the effect of manipulating BNA2 on lifespan and carried out these analyses in both short- and long-lived strains (Figure 10). We also similarly examined GLN1. Interestingly, expression of these genes in long-lived strains shortened lifespan, whereas in short-lived strains, the lifespan was typically increased or not affected. This finding supports the idea of disparate lifespan affecting mechanisms in wild isolates and their differences with those of lab strains. We also added figures on various metabolic pathways and heavily revised the manuscript.

Regarding, BNA2 a recent study showed that its overexpression extends life-span (Beas AO, Nat Commun, 2020), thus the opposite of what could be expected based on this report. Furthermore, BNA2 overexpression and increased activity in the kynurenine pathway affected life-span without impacting on NAD+ levels, but through an as of yet poorly understood mechanism increasing the levels of kynurenine branch point metabolites e.g. xanthurenate (Beas AO, Nat Commun, 2020). Notably, the study furthermore identified age-related accumulation of lipid droplets in yeast, that were inhibited by BNA2 overexpression, an observation that may relate to the positive correlation between lipid metabolic process and replicative longevity in this study and its anti-correlation to kynurenine pathway metabolites. In fact the manuscript at hand supports the importance of kynurenine pathway metabolites independently of NAD+, since NAD+ levels do not correlate with longevity, yet the authors focus all their discussion of mechanism around NAD+ and histone deacetylases. The observation that NADH levels correlate with replicative longevity is brushed away with a discussion about technical problems in assessing their levels faithfully in vivo (p14, lines 367-375).

Thank you for these insightful comments. As mentioned above, we paid a particular attention to BNA2 and the kynurenine pathway. We tested the overexpression of BNA2 in long- and short-lived strains and discuss the findings in the revised paper. We cited previous papers and also discuss relevance of these findings to our study.

Reviewer #3:[…] – One of my main concerns is the strategy that was used by the author in order to dissect this given trait, which is longevity. It is probably fair to try to find some correlations between different datasets (even if there are not totally related) but not sure the conclusions should be trusted without additional experimental validation. As clearly mentioned in the 'Material and Methods' section, cells were grown for 2 days at 30°C for the RLS assay. By contrast, strains were collected at OD = 0.4 for the RNAseq and metabolite profiling. Obviously, this makes a serious difference. Again, it's fair to compare the datasets but not sure, general conclusions can be drawn without experimental validations.

Thank you for the comment. We performed additional experiments and included an in-depth analysis of the data. With regard to culturing conditions, we would like to clarify the experimental procedure. Cells were maintained for 2 days at 30^0^C prior to the RLS assay. It takes 2 days starting from frozen cells to fully grow the culture. After 2 days, cells were in the form of single colonies from culture plates, and these were transferred to new plates for subsequent RLS assays. In these conditions, cells were always exposed to constant glucose levels. This condition is very similar to the situation when cells were grown until the OD=0.4. We revised the Methods section to clarify the experimental procedure.

– Line 85'thus, their natural lifespan variation must be encoded in their respective genomes'As the authors explored the different traits with no-environmental changes in the lab, this is a very strange conclusion/statement.

Thank you for the comment. We also would like to clarify this statement. In this sentence, we suggest that these wild isolates have been exposed to different niche-specific conditions prior to bringing them to the lab, whereas they were cultured under the same conditions for all strains in the lab. The accumulated genetic changes cause phenotypic variation which includes lifespan. We hope that with the revisions, this concept is more clear.

– Line 107'There was also a group of pathogenic *S. cerevisiae* strains isolated from immunocompromised patients'*S. cerevisiae* is not a pathogen. These strains are called 'clinical isolates'.

Thank you for commenting on this issue. We replaced the sentence as follows: “'There was also a group of clinical isolates; *S. cerevisiae* strains isolated from immunocompromised patients”.

– Line 109 – 111'To understand the forces that shape diversity of lifespan and the associated mechanisms, we first sequenced and assembled the genomes of these 76 isolates. Consistent with a previously published report [49], we found substantial genetic variation among these strains (~655,000SNPs).' The genome of more than 2,000 *S. cerevisiae* isolates were now sequenced. Were these isolates not sequenced yet. The authors mentioned the consistency of their results with a study that was published 12 years ago and dealing with Sanger sequenced genome at a coverage of around 3x. How can that be consistent? There is a quantum leap in data quality and precision there… no real comparison seems possible and insightful. Much of the yeast population genomic literature is missing here!

Thank you for the comment. We agree and removed the section discussing the findings from genome sequencing. We plan to perform a more comprehensive analysis of genome sequencing data (for both those we sequenced and other strains) in future studies.

– Surprisingly, one study that explored for the first time intra- and interspecies RLS variation in yeast is not cited?!?!? Jung et al. 2015 PLoS one.

Thank you. We updated our cited literature and included this citation.

– Line 220'The resulting topology was largely consistent with their phylogeny with a clear separation between *S. cerevisiae* and *S. paradoxus* strains (Figure 2A).'What does 'largely consistent' mean? At the intra-specific level (*S. cerevisiae*), the topology is not consistent at all with the genomic data.

Thank you for this helpful comment and bringing this issue to our attention. We replaced the text as follows: “The resulting topology of species specific separation was largely consistent with a clear separation between *S. cerevisiae* and *S. paradoxus* strains (Figure 2A), however, at the intra-specific level, the topology was not consistent with the phylograms based on the genomic data**.”**

– The ‘gene expression and metabolite’…' paragraph is too wordy and poorly structured. It probably needs to be rewritten.

Thank you. This section was rewritten.

– The conclusion begins with a lengthy commentary on laboratory strain S288c. This part does not add anything and seems a little off topic.

Thank you. We have significantly revised our discussion.